# Relationships among creativity indices: Creative potential, production, achievement, and beliefs about own creative personality

**Chiaki Ishiguro**[1], **Yuki Sato**[2,3], **Ai Takahashi**[4], **Yuko Abe**[4], **Etsuko Kato**[4], **Haruto Takagishi** [ID][3,5]*

**1** College of Informatics and Human Communication, Kanazawa Institute of Technology, Ishikawa, Japan, **2** College of Liberal and Sciences, Tamagawa University, Tokyo, Japan, **3** Graduate School of Brain Sciences, Tamagawa University, Tokyo, Japan, **4** College of Arts, Tamagawa University, Tokyo, Japan, **5** Brain Science Institute, Tamagawa University, Tokyo, Japan

* haruharry@gmail.com

**Data Availability Statement:** All relevant data are within the paper and its Supporting information files.

## Abstract

Although creativity has been measured in various ways (ideas, products, achievements, and personality), the relationships between these measurements remain unclear. The current study examines whether divergent thinking predicts creative behavior (i.e., creative production and achievement) and whether beliefs about own creative personality influences the link between divergent thinking and creative behavior. Eighty-eight undergraduate students were assessed via a divergent thinking test, a creative production test, and a creative achievement questionnaire. The results showed that divergent thinking was positively associated with both creative behaviors (i.e., creative production in fine arts and achievement). In addition, beliefs about own creative personality moderated the relationship between divergent thinking and creative achievements, in that this relationship was stronger when Creative Personality Scale scores were higher. The current findings suggest some associations among creativity indices: divergent thinking promotes creative achievements, and this relation is moderated by beliefs about own creative personality. Further investigation is required to specify the causal relationships among creativity indices.

## Introduction

Creativity is a unique human ability that psychological researchers have claimed is demonstrated by each individual in their everyday lives [1, 2]. However, some fundamental questions regarding creativity remain to be solved, including what creativity is and how this construct can be measured [3, 4]. The most popular and consensual definition of creativity describes it with just two words, "novelty" and "useful," meaning that creative products should be original and valuable [2, 3]. Based on this simple definition, researchers have classified the types of creativity according to various approaches ranging from cognitive and personality approaches to systems and sociocultural theories [5–9]. For instance, the 4 Ps theory claims that creativity could be attributed to the Person (individuals' cognitive and metacognitive traits), Process

**Funding:** This work was supported by MEXT Promotion of Distinctive Joint Research Center Program Grant Number JPMXP0621467950. The funders had no role in study design, data collection and analysis, decision to publish, or preparation of the manuscript.

**Competing interests:** The authors have declared that no competing interests exist.

(psychological process to perform creative acts), Press (environment and situations that produce creativity), and Product (specific works and achievement to demonstrate creativity) [10]. Eventually, this model was further developed into an integrated model with additional factors [11].

Based on these categorizations, many researchers have developed various instruments to measure creativity, such as scales for assessing the Person, Process, and Product dimensions of the 4 Ps theory. In this theory, the Person dimension refers to individual traits that correlate with creative individuals, such as creative personality, attitudes, and motivation [12]. The Process dimension refers to psychological processes pertaining to creative thinking and activities, such as divergent thinking (DT) [13]. The Product dimension refers to creative process outcomes, such as drawings, poems, inventions, among others.

When focusing on these 3 Ps, some of the traits and abilities that are classified into the Person and Process dimension are referred to as creative potential [14, 15]. Creative potential, in turn, is expected to relate to and predict creative production and achievements, which relate to the Product dimension [15, 16]. Following these theoretical assumptions, various researchers have examined the impact of creative personality (Person dimension) and creative thinking (Process dimension) on creative production and achievements (Product dimension). However, the relationships among these creativity indices remain unclear.

## The role of divergent thinking on creative performance

Among these 3 Ps, the relationship between the Process and Product dimensions has been extensively researched. DT is a representative creative potential in the Process component [14], which is typically tested through idea generation tasks. Representative tests include the Torrance Test of Creative Thinking (TTCT) [17], the tests of Wallach and Kogan [18], and those proposed by Williams [19] and by Guilford [11], among others.

Psychologists have examined the impact of DT on the Product dimension by assessing creative production and achievements. When estimating individuals' creative production, researchers focus on a particular creative field (e.g., fine arts or writing), request participants to produce a creative work in that field, and then ask other participants to evaluate the products. The Consensual Assessment Technique (CAT) is usually applied in this type of evaluation [20]. In it, first, researchers ask participants to produce creative works of a specific type. Next, they ask experts to rate these products according to their specialty [20]. This procedure is time-consuming and taxing, yet it is a valid, objective way to measure an individual's creative production in target creative domains.

Kaufman and Beghetto [9] implied that individuals with high DT could make more innovative products. Baer [21] examined the relationship between fluency in a verbal DT test and creative production (e.g., poetry, story writing, word problems, and equations) with a sample comprising participants from different age groups (children to adults). Although the findings identified positive and weak to moderate relationships between verbal fluency and story writing in a relatively small sample of children (N = 50), this significant association was not replicated in other samples of children and adults. Baer [21–23] claimed that the actual creative production or performance should be considered when examining DT and behavior. However, since studies on the relationship between DT and actual creative production are scarce, it is unclear whether DT relates to actual creative production just as the finding of Baer [21–23].

Meanwhile, the impact of DT on creative achievements has been examined in lots of psychological research; however, the evidence on their association is not consistent, and it was generally positive and weak in most studies, which were conducted with samples containing a wide range of age groups, from children to adults [17, 24–28]. Most studies measured creative

achievements using self-report questionnaires, such as the Creative Activity and Accomplishment Checklist (CAAC) [29] and the Creative Achievement Questionnaire (CAQ) [30]. These questionnaires can easily measure individuals' overall creative achievement scores or scores for specific creative domains. For instance, Wallach and Wing [26] showed a positive association between DT and creative achievements; furthermore, in their experimental survey with 503 first-year university students, they showed that DT predicted non-academic outcomes that IQ and standardized tests alone did not.

Subsequent longer-term longitudinal studies have examined the predictive power of DT in more detail. Kogan and Panvoke [24] showed that children's DTs correlated with their creative achievements in the fifth but not in the 10th grade. An 18-year follow-up study also showed that the DT of high school students was only associated with specific areas of creative achievements in their adult lives [25]. Torrance [17] designed the Torrance Test of Creative Thinking (TTCT) to indicate that DT promotes creative achievements. In addition, previous empirical research with a 40- or 50-year follow-up found that creative ideation scores predict future creative achievement [31–33]. Further, it is suggested in a meta-analysis [34] and follow-up study [31] that DT has a better predictive association with creative achievements than IQ. At the same time, some researchers have argued that the link between DT and creative achievement is weak ($r$ score is less than .30), and sometimes insignificant [35].

## Individuals' beliefs about own creative personality mediates the link between DT and creative behavior

Here, a question is begged: what can explain these findings on the inconsistent and weak association between DT and creative production and achievements? A recent theory by Karwowski and Beghetto [36] provides an outlook on this matter, proposing that individuals actualize their creative potential based on their personal intentions. They named the theory Creative Behavior as an Agentic Action (CBAA), positing that creative self-beliefs (which encompasses creative self-efficacy and perceived value of creativity) is a personal factor that mediates (through creative self-efficacy) and moderates (through perceived value of creativity) the relationship between DT and creative behavior [36, 37]. Hence, CBAA proposes that individual traits and attitudes toward creativity can influence the relationship between DT and creative behavior. Importantly, these propositions of CBAA raise the question as to whether other characteristics at the individual level can influence this relationship.

Kaufman [38] suggested that self-assessed measures of creative personality (e.g., openness to experience and beliefs about own creative personality) can be used to perceive how individuals view their creativity in addition to measures of creative self-beliefs (e.g., creative self-efficacy and perceived value of creativity). That is, beliefs about own creative personality can have a mediating or moderating role on the association of DT with creative production and achievement. Since these trait-related measures of creativity are assumed to be conative dimensions of creative potential [14], they cannot be preceded by DT, namely, they cannot have a mediating role on the relation between DT and creative behavior. Here, an interrogation arises: can beliefs about own creative personality strengthen individuals' actualization of DT into creative behaviors? The confirmation of such an assumption may provide support to the triangle relationship of the Person, Process, and Product dimensions.

## Research objective

Considering the aforementioned problems, this study examines (1) whether DT associates with creative production and achievements, and (2) how beliefs about own creative personality moderate the associations among creativity potential, production, and achievement.

In the present study, an experiment comprising creative production tasks (rated using CAT and CAQ) was conducted to measure DT and creative production and achievement (by applying the S-A Creativity test and using the TTCT for Japanese speakers). In addition, researchers collected data on participants' beliefs about own creative personality to examine the relationships among creativity indices. We used the self-assessed Creative Personality Scale (CPS) [39, 40] to examine beliefs about own creative personality because it comprises items that relate to various creative traits of thinking, problem-solving, and imagination. Although the CPS does not specifically assess personality, it does yield self-assessed data on a wide range of creative traits, allowing us to understand the moderating role of beliefs about own creative personality on the dimensions of creativity.

While the S-A Creativity test measures DT and the CAQ measures total achievements in various creative domains, the creative production tasks focused on participants' creative production ability in a specific creative domain. We expected that the assessments conducted related to the creative production task would cover as much of the related domain as possible. However, given that creative production tasks in a specific area require significant engagement from the participants, potentially causing fatigue, it was challenging to conduct a series of creative production tasks. Accordingly, and considering the suggestions of prior research on the association of the creative domains of fine arts and writing with innovative products and achievements [21, 25, 26, 28], our creative production task included these two creative domains. Although the number of creative domains we have explored is limited, our findings shed light on the relationship between the 3 Ps—with the Person domain being represented by beliefs about own creative personality, the Process domain being represented by DT, and the Product domain being represented by creative production quality and creative achievements quantity—by simultaneously measuring the three measures of creativity in the same sample.

## Materials and methods

### Participants

DT tests (e.g., S-A Creativity test) can be administered across diverse populations, from kindergarten to adulthood. Because this study included creative achievement and creative tasks, we recruited participants with the ability and experience to perform them. Participants were recruited through postings at the authors' research institution. 88 undergraduate students from a Japanese university (male = 44, female = 44, mean age = 19.4, standard deviation = 2.3) participated in the study (S1 Dataset). Each participant received 5,000 JPY (US$44.20) as a token of appreciation. This experiment was approved by the ethics committee of Tamagawa University (Tokyo, Japan; approval no. TRE18-010), and was conducted in accordance with approved guidelines. Participants were informed with oral and written instructions, and they signed consent forms before participating in the study.

### Procedure

The experiment included four types of creativity measurements: a creative thinking test, a creative production test, a creative achievement questionnaire, and a creative personality scale. The experiment was conducted in a large lecture room. Upon entering, participants were seated at tables at least one seat away from each other. The whole procedure lasted approximately three hours, and participants took breaks of about five minutes between each test and task.

## DT

DT was measured using the S-A Creativity test standardized for Japanese participants [41]. Guilford supervised the development of this instrument, which is used to assess DT based on the TTCT; further it has shown high reliability and validity and can be administered from fourth grade to college students [41]. Various researchers have used the S-A Creativity test in research on creativity in Japan; one example is a study showing positive associations between personality and problem-solving in daily life [42], and two others examined the relationship between biological and physiological data [43, 44].

The test comprises three types of subtests: Unusual Uses Task, Product Improvement, and Just Suppose in TTCT. The Unusual Uses Task requires participants to generate unique ways of using typical objects, such as a newspaper, button, and chopsticks. For example, participants were asked the following question: "Other than reading, what other uses are there for newspapers? For instance, you may use them to wrap things." The Product Improvement subtest requires participants to imagine different ways for improving a product, such as a TV, pot, or desk. For instance, participants were asked, "What kind of TV can you imagine having? For example, a TV that can show 3D images. Write down as many as possible." Finally, the Just Suppose subtest requires participants to imagine the consequences when something unimaginable happens, such as when there are no clocks in the world. For example, participants were asked, "What would happen if all the mice in the world disappeared? One possible consequence is that cats might become hungry." Participants are given two minutes for a trial and five minutes for two questions in each subtest.

The test is scored based on four aspects: fluency, flexibility, originality, and elaboration. Fluency is a measure of the ability to generate many ideas, being evaluated by the number of responses excluding those that are inappropriate or difficult to interpret. Flexibility is the ability to generate ideas from a wide range of perspectives, being assessed by the number of categories in the ideas generated and according to a criterion table (i.e., indicating a particular answer's classification and how many points it receives) or equivalent judgment [41]. Originality is the ability to generate ideas different from those of others, which is again evaluated based on a criterion table (i.e., indicating the frequency of occurrence of categories for each response) [41]. Those with a frequency of occurrence of less than 1% scored 2 points, of 1–5% scored 1 point, and of more than 1% scored 0 points [41]. Finally, elaboration is the ability to think concretely about ideas, being evaluated based on the total number of responses weighted by a criterion table (i.e., describing how well the response depicts the means and structure of the purpose or functions) or equivalent judgments [41]. Torrance [45] initially established these four dimensions for evaluating ideas based on the elements of DT as proposed by Guilford [13].

One trained judge from an external professional organization performed all of the ratings; the expert judge evaluated the participants' responses while following the scoring charts developed according to data from a sample of the Japanese population (Tokyo Shinri Corporations) [41].

## Creative production

Creative production was measured using tasks requiring participants to create original artwork, such as cutouts and Haiku pieces, as proposed by Amabile [20], who conducted the CAT using tasks like collage-making and poetry. Haiku is a popular Japanese poetic form, which involves a 17-syllable verse form comprising three metrical units of 5, 7, and 5 syllables that must have a seasonal word called "Kigo." The first task was to compose Haiku pieces

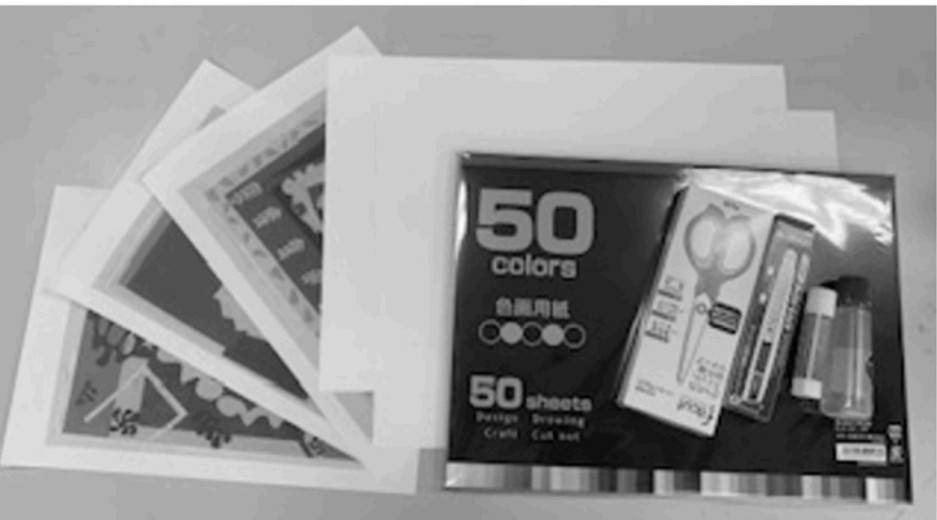

**Fig 1. Materials for the creative production task in fine arts.**

using Kigo; participants were asked to create three original Haikus featuring two themes ("crocus" and "spring breeze") in 30 minutes.

The second task was to produce original cutout pieces for the given themes of "heart" and "the circus" in 30 minutes. A researcher provided each participant with a piece of A3-sized drawing paper, a book with 50 different-colored papers, stick glue and starch paste, a cutter, scissors, and a pencil (Fig 1). Participants could cut or tear the colored paper with scissors or their hands and paste it on the drawing paper (Fig 2). To guide their understanding of the task,

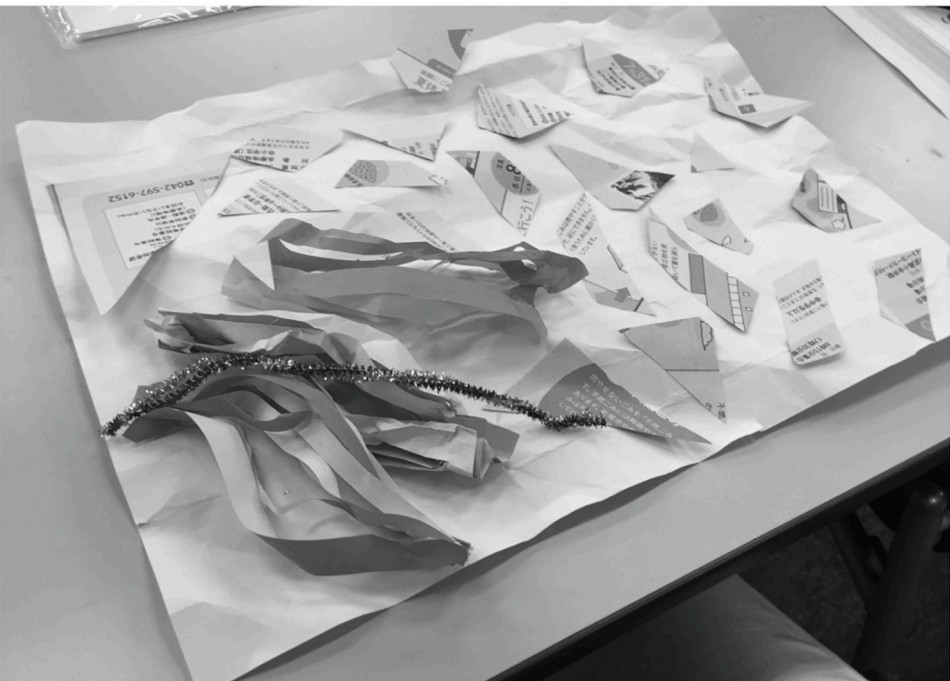

**Fig 2. Sample cutout in the creative production task.** The first author created this work for her preliminary experiment where she did a similar task on the theme of "water".

we provided three sample cutouts (images of *Pierrot's Funeral*, *The Codomas*, and *The Wolf* by Henri Matisse, printed on copier paper) without the titles or the artists' names.

## Rating product creativity

The Haiku pieces were rated using the CAT [20] by five expert judges and teachers in Haiku and language. They were asked to evaluate the pieces with information composed by the undergraduates with the themes of "crocus" and "spring breeze" in 30 minutes. The judges rated the Haiku pieces on 10 evaluative dimensions (understanding Kigo, creativity of expression, novelty of ideas, expression intention, sophistication, rhythm, appropriate style, correctness of format, and likeness). In addition, the pieces were rated relative to one another from 1 (*very low*) to 5 (*very high*). Following Amabile's [20] procedure, factor analyses were performed for rating three Haikus of each participant by five raters. The eigen values (4.15, 1.08, 0.19, -0.02) suggested that the dimensions consisted of two factors and the reliabilities as a consistency were good ($\alpha$s > .80). Further, the CAT scoring procedure requires inter-rater agreement [46, 47]; although the current study calculated ICC (2, k) using the "irr" package for R [48] based on a mean-rating (k = 5), absolute-agreement, and 2-way mixed-effects model, the inter-rater agreements were quite poor (ICCs (2, k) < .20). Thus, the results of the Haiku ratings were not included for further analysis.

The cutout pieces were also rated using the CAT [20] by five expert judges and teachers in fine arts, art education, and art history. They were asked to evaluate the pieces with the information that the undergraduates had created using the themes of "heart" and "the circus" in 30 minutes. The judges rated the pieces on 10 evaluative dimensions: novelty in material use, novelty of thematic ideas, variety of forms, attention, consideration of composition, ordonnance, use of colors, expressiveness, aesthetic appeal, and likeness. In addition, the pieces were rated relative to one another from 1 (*very low*) to 5 (*very high*). Factor analyses were performed for rating two products by each participant by five raters. The eigen values (5.72, 0.46, 0.21, 0.08) suggested that the dimensions consisted of one factor and the reliability as a consistency was good ($\alpha$ = .95). Further, the CAT scoring procedure requires the inter-rater agreement [46, 47]; the current study calculated ICC (2, k) into 0.75 [CI: 0.62–0.84] based on a mean-rating (k = 5), absolute-agreement, and 2-way mixed-effects model, which was an acceptable level of agreement.

## Creative achievement and beliefs about own creative personality

A series of self-report questionnaires were used to measure the participants' creative achievement and beliefs about own creative personality. The first questionnaire was the CAQ by Carson et al. [30], which measures achievement across 10 creative domains (fine arts, music, dance, architectural design, writing, comedy, invention, science, theater and film, and cooking). Total achievement scores in all domains were used in the analysis.

The second was the CPS [40], which consists of 20 items selected from the International Personality Item Pool [49, 50], which is a 5-point Likert scale (1: *not at all* to 5: *very much*) and measures creativity in general and in specific domains (e.g., science, interpersonal communication, writing, art). In this study, the overall mean was used for the analysis.

## Sensitivity power analysis

This study did not use a prior sample design. Therefore, we conducted a sensitivity power analysis using G*Power 3.1 [51]. For a correlation analysis between two variables with power set at 0.8, we could theoretically detect an effect size ($\rho$) greater than 0.289 with 88 participants, and greater than 0.291 with 87 participants.

Regarding moderation effect analyses, with power set at 0.8, we could theoretically detect an effect size ($f^2$) greater than 0.091 with 88 participants, and an effect size ($f^2$) greater than 0.092 with 87 participants.

## Results

The descriptive statistics of the S-A Creativity test, creative production (in fine arts), total score of the CAQ, and the CPS score are shown in Table 1. The distribution of scores for each task is illustrated in Fig 3. A Shapiro-Wilk test was used to test whether scores on each measure of creativity were normally distributed. Fluency ($W = 0.97$, $p = .053$), flexibility ($W = 0.98$, $p = .102$), originality ($W = 0.98$, $p = .402$), elaboration ($W = 0.99$, $p = .813$), creative production ($W = 0.98$, $p = .180$), and CPS score ($W = 0.98$, $p = .196$) were normally distributed, but the creative achievement score was not ($W = 0.77$, $p < 0.0001$). Therefore, log-transformed values of the creative achievement scores were used in the analysis.

The scores for fluency and flexibility highly correlated with those for elaboration in the past surveys among Japanese people. Therefore, the total creativity score was defined as the sum of the raw score of originality and elaboration in the S-A Creativity test. However, Guilford [13] viewed DT as multidimensional, and Torrance [45] discouraged researchers from using the total score of the four aspects. Since the use of the total score of the TTCT is controversial and reporting of all four subscales is encouraged [52], we first assessed the reliability (Cronbach's alpha) for each subscale in the current sample and then examined the relationships among subscales. A reliability score greater than .80—or .70 in psychometric assessment—is required to avoid regression dilution and reduction in testing power [53, 54].

The reliability of fluency (.80) and elaboration (.77) were adequate, whereas those of other evaluation aspects were not (flexibility: .67; originality: .56). Furthermore, the correlation between fluency and elaboration was relatively high ($r_{(86)} = .889$, $p < .0001$). Therefore, as the subsequent analyses could not include S-A Creativity test subscales simultaneously, the scores for fluency and elaboration were used individually because these were the most reliable, and such subscales demonstrated similar characteristics to the others. The results of the analysis of flexibility and originality are not mentioned in the text due to their low reliability and can be checked in the Supporting Information (S1–S4 Tables).

While the S-A Creativity test scores (fluency and elaboration) were positively correlated with the creative production score (fluency: $r_{(85)} = .258$, $p = .016$, elaboration: $r_{(85)} = .309$, $p = .004$) and the CAQ score (fluency: $r_{(86)} = .306$, $p = .004$, elaboration: $r_{(86)} = .230$, $p = .031$), the creative production score was not correlated with the CAQ score ($r_{(85)} = .118$, $p = .277$) (Table 2 and Fig 4). Additionally, while the CPS score was positively correlated with the CAQ score ($r_{(85)} = .230$, $p = .032$), the CPS score was not correlated with the S-A Creativity test

**Table 1. Descriptive statistics of the creativity indices.**

| Variables | n | Mean | Median | SD | Min | Max |
|---|---|---|---|---|---|---|
| S-A Creativity test (fluency) | 88 | 31.14 | 30 | 8.59 | 7 | 49 |
| S-A Creativity test (flexibility) | 88 | 22.01 | 22 | 4.98 | 6 | 33 |
| S-A Creativity test (originality) | 88 | 8.81 | 8 | 4.16 | 0 | 20 |
| S-A Creativity test (elaboration) | 88 | 24.75 | 25 | 7.87 | 5 | 45 |
| Creative production (fine arts) | 87 | 3.27 | 3.25 | 0.54 | 2.3 | 4.5 |
| CAQ total | 88 | 6.64 | 4.5 | 7.24 | 0 | 35 |
| Creative personality scale | 87 | 3.37 | 3.368 | 0.42 | 2.316 | 4.211 |

CAQ = creative achievement questionnaire

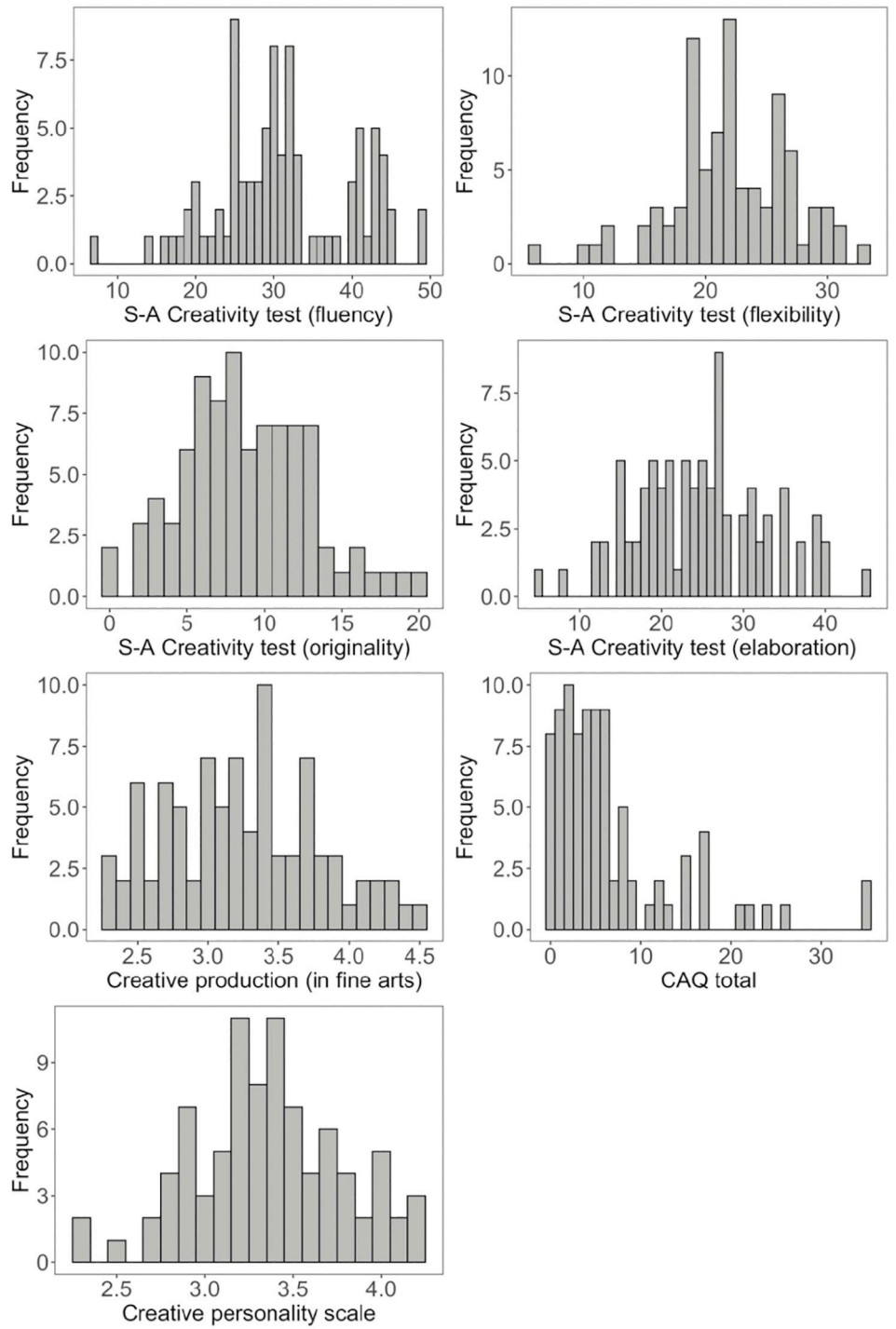

**Fig 3. Distribution of scores among creativity indices.**

scores (fluency: $r_{(85)} = .104$, $p = .336$, elaboration: $r_{(85)} = .022$, $p = .838$) nor the creative production score ($r_{(84)} = .189$, $p = .081$).

Next, we examined the moderating effect of the CPS score on the association between S-A Creativity test (fluency) and creative behavior. First, a multiple regression analysis was

**Table 2. Correlations of creativity indices.**

|  |  | 1 | 2 | 3 | 4 | 5 | 6 | 7 | 8 |
|---|---|---|---|---|---|---|---|---|---|
| 1 | S-A Creativity test (fluency) |  |  |  |  |  |  |  |  |
| 2 | S-A Creativity test (flexibility) | 0.875** |  |  |  |  |  |  |  |
| 3 | S-A Creativity test (originality) | 0.746** | 0.794** |  |  |  |  |  |  |
| 4 | S-A Creativity test (elaboration) | 0.889** | 0.791** | 0.701** |  |  |  |  |  |
| 5 | CAQ total | 0.276** | 0.222* | 0.155 | 0.221* |  |  |  |  |
| 6 | CAQ total(log) | 0.306** | 0.231* | 0.174 | 0.239* | 0.875** |  |  |  |
| 7 | Creative production (fine arts) | 0.258* | 0.323** | 0.254* | 0.309** | 0.073 | 0.118 |  |  |
| 8 | Creative personality scale | 0.104 | 0.181 | 0.119 | 0.022 | 0.257* | 0.230* | 0.189 |  |

Note:

** $p < .01$,

* $p < .05$

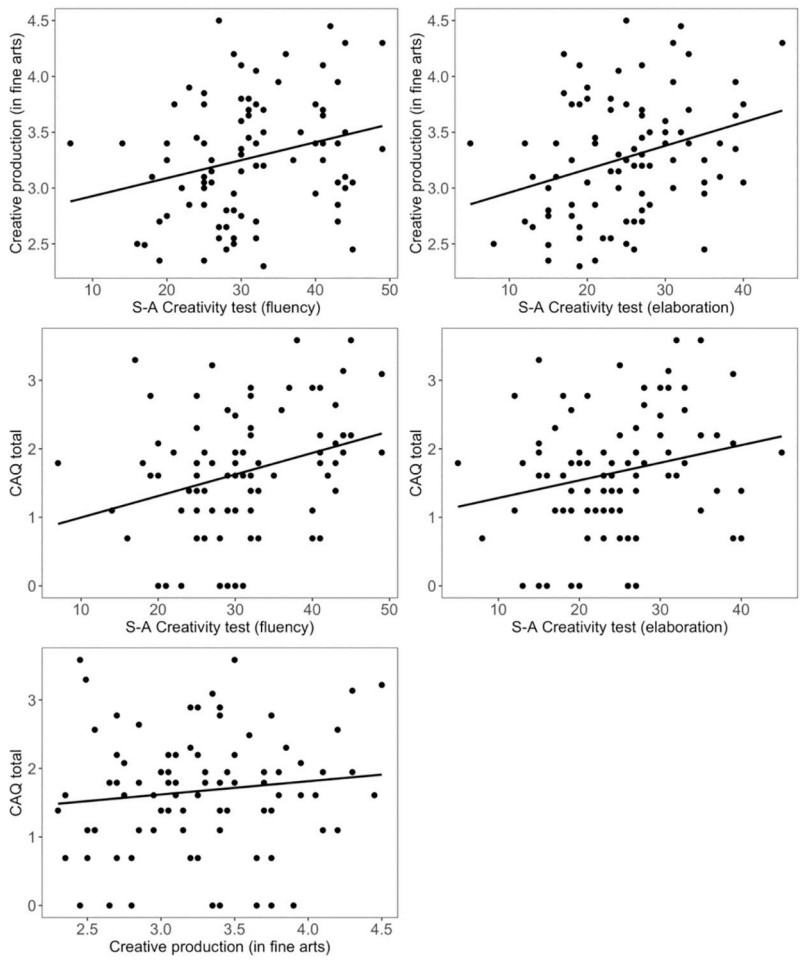

**Fig 4. Results of correlational analysis among creativity indices.**

**Table 3. Results of multiple regression analysis in the effect of fluency with creative production score as the objective variable.**

| Predictors | $b$ | 95% CI | | $T$ | df | $p$ | $\beta$ |
|---|---|---|---|---|---|---|---|
| | | LL | UL | | | | |
| Step 1 ($R^2$ = .065, $p$ = .018) | | | | | | | |
| Intercept | 3.27 | 3.16 | 3.38 | 57.7 | 84 | < .0001 | |
| S-A Creativity test (fluency) | 0.02 | 0.003 | 0.029 | 2.4 | 84 | .018 | .26 |
| Step 2 ($R^2$ = .091, $p$ = .019) | | | | | | | |
| Intercept | 3.27 | 3.16 | 3.38 | 58.2 | 83 | < .0001 | |
| S-A Creativity test (fluency) | 0.01 | 0.002 | 0.028 | 2.3 | 83 | .027 | .24 |
| CPS | 0.21 | -0.06 | 0.48 | 1.5 | 83 | .125 | .16 |
| Step 3 ($R^2$ = .118, $p$ = .015) | | | | | | | |
| Intercept | 3.28 | 3.17 | 3.39 | 58.5 | 82 | < .0001 | |
| S-A Creativity test (fluency) | 0.01 | 0.001 | 0.027 | 2.1 | 82 | .039 | .22 |
| CPS | 0.24 | -0.03 | 0.5 | 1.7 | 82 | .084 | .18 |
| S-A Creativity test (fluency) × CPS | -0.03 | -0.06 | 0.01 | -1.6 | 82 | .117 | -.17 |

CI: confidential interval, LL: lower limits, UL: upper limits, CPS: creativity personality scale

conducted with creative production as the objective variable and the S-A Creativity test score (fluency), CPQ score, and the interaction term between the S-A Creativity test score (fluency) and CPQ score as explanatory variables (Table 3). In Step 1, we examined the main effect of the S-A Creativity test score (fluency) on creative production, finding a positive effect ($\beta$ = .26, $t_{(84)}$ = 2.4, $p$ = .018).

In Step 2, the CPS score was added to the explanatory variables, and although the effect of S-A Creativity test score (fluency) remained ($\beta$ = .24, $t_{(83)}$ = 2.3, $p$ = .027), that of the CPS score was not observed ($\beta$ = .16, $t_{(83)}$ = 1.5, $p$ = .125).

In Step 3, the interaction term between the S-A Creativity test score (fluency) and the CPS score was added to the explanatory variables. Still, there was no interaction effect between these two variables ($\beta$ = -.17, $t_{(82)}$ = -1.6, $p$ = .117).

In addition, a multiple regression analysis was conducted with the CAQ score as the objective variable, S-A Creativity test (fluency) and CPS scores as the explanatory variables, and the interaction term between the S-A Creativity test score (fluency) and the CPS score (Table 4).

In Step 1, we examined the main effects of the S-A Creativity test score (fluency) on the CAQ, finding a positive effect ($\beta$ = .31, $t_{(85)}$ = 3.0, $p$ = .004). In Step 2, the CPS score was added as an explanatory variable, again showing the positive effect of the S-A Creativity test score (fluency) ($\beta$ = .29, $t_{(84)}$ = 2.8, $p$ = .006), but not that of the CPS score ($\beta$ = .20, $t_{(84)}$ = 2.0, $p$ = .053). In Step 3, the interaction terms of the S-A Creativity test score (fluency) and the CPS score were added to the explanatory variables, showing an interaction effect between these two variables ($\beta$ = .33, $t_{(83)}$ = 3.5, $p$ = .001, $\Delta R^2$ = .110).

The simple main effect test results showed that the S-A Creativity test score (fluency) had a positive effect on the CAQ score in the high CPS group ($\beta$ = .68, $t_{(83)}$ = 4.6, $p$ < .0001, Fig 5A), but not in the low CPS group ($\beta$ = -.04, $t_{(83)}$ = 0.3, $p$ = .767).

We found similar results using the S-A Creativity test score (elaboration). A multiple regression analysis was conducted using creative production as the objective variable and S-A Creativity test (elaboration), CPS score, and the interaction term between S-A Creativity test (elaboration) and CPS score as explanatory variables (Table 5).

In Step 1, we found that the S-A Creativity test (elaboration) had a positive main effect on creative production ($\beta$ = .31, $t_{(84)}$ = 3.0, $p$ = .004). In Step 2, the CPS score was included as an

**Table 4. Results of multiple regression analysis in the effect of fluency with creative achievement score as the objective variable.**

| Predictors | $b$ | 95% CI | | $T$ | df | $p$ | $\beta$ |
|---|---|---|---|---|---|---|---|
| | | LL | UL | | | | |
| Step 1 ($R^2$ = .095, $p$ = .004) | | | | | | | |
| Intercept | 1.66 | 1.48 | 1.84 | 18.2 | 85 | < .0001 | |
| S-A Creativity test (fluency) | 0.03 | 0.01 | 0.05 | 3.0 | 85 | .004 | .31 |
| Step 2 ($R^2$ = .135, $p$ = .002) | | | | | | | |
| Intercept | 1.66 | 1.48 | 1.84 | 18.5 | 84 | < .0001 | |
| S-A Creativity test (fluency) | 0.03 | 0.009 | 0.05 | 2.8 | 84 | .006 | .29 |
| CPS | 0.42 | -0.005 | 0.85 | 2.0 | 84 | .053 | .20 |
| Step 3 ($R^2$ = .245, $p$ < .0001) | | | | | | | |
| Intercept | 1.62 | 1.46 | 1.79 | 19.2 | 83 | < .0001 | |
| S-A Creativity test (fluency) | 0.03 | 0.013 | 0.053 | 3.3 | 83 | .001 | .32 |
| CPS | 0.34 | -0.065 | 0.744 | 1.7 | 83 | .099 | .16 |
| S-A Creativity test (fluency) × CPS | 0.09 | 0.038 | 0.139 | 3.5 | 83 | .001 | .33 |

CI: confidential interval, LL: lower limits, UL: upper limits, CPS: creativity personality scale

explanatory variable, and although the effect of S-A Creativity test (elaboration) remained ($\beta$ = .30, $t_{(83)}$ = 2.9, $p$ = .004), that of the CPS score was not observed ($\beta$ = .18, $t_{(83)}$ = 1.8, $p$ = .080). In Step 3, the interaction term between the S-A Creativity test (elaboration) and the CPS score was included as an explanatory variable. However, there was no interaction effect between the two ($\beta$ = -.11, $t_{(82)}$ = -1.1, $p$ = .294).

Further, a multiple regression analysis was conducted with CAQ score as the objective variable and S-A Creativity test (elaboration), CPS score, and the interaction term between the S-A Creativity test (elaboration) and CPS score as explanatory variables (Table 6).

In Step 1, we found that S-A Creativity test (elaboration) had a positive main effect on the CAQ score ($\beta$ = .23, $t_{(85)}$ = 2.2, $p$ = .031). In Step 2, the CPS score was included as an explanatory variable, and both S-A Creativity test (elaboration) ($\beta$ = .23, $t_{(84)}$ = 2.2, $p$ = .031) and the CPS score ($\beta$ = .23, $t_{(84)}$ = 2.2, $p$ = .032) showed a positive effect. In Step 3, the interaction term of S-A Creativity test (elaboration) and the CPS score was included as an explanatory variable, showing the interaction effect between the two ($\beta$ = .39, $t_{(83)}$ = 4.0, $p$ < .0001).

The simple main effect test results showed that S-A Creativity test (elaboration) had a positive effect on the CAQ score in the high CPS group (β = .70, $t_{(83)}$ = 4.6, $p$ < .0001, Fig 5B), but not in the low CPS group ($\beta$ = -.09, $t(83)$ = -0.7, $p$ = .482).

## Discussion

This study examined whether DT associates with creative production and achievements. The results confirmed these associations, supporting the theoretical assumption by Kaufman and Beghetto on DT and creative behavior [9]. We also examined whether beliefs about own creative personality moderated the associations among DT and creative production and achievements. The results of the multiple regression analysis revealed that the creative personality score moderated the relationship between DT (fluency and elaboration) and creative achievements. That is, creative achievement scores rose when DT scores were higher, signifying a more robust relationship when individuals believed themselves to have a more creative personality. The demonstration of this moderating effect configures a novel finding that had been left unclear in previous research [55].

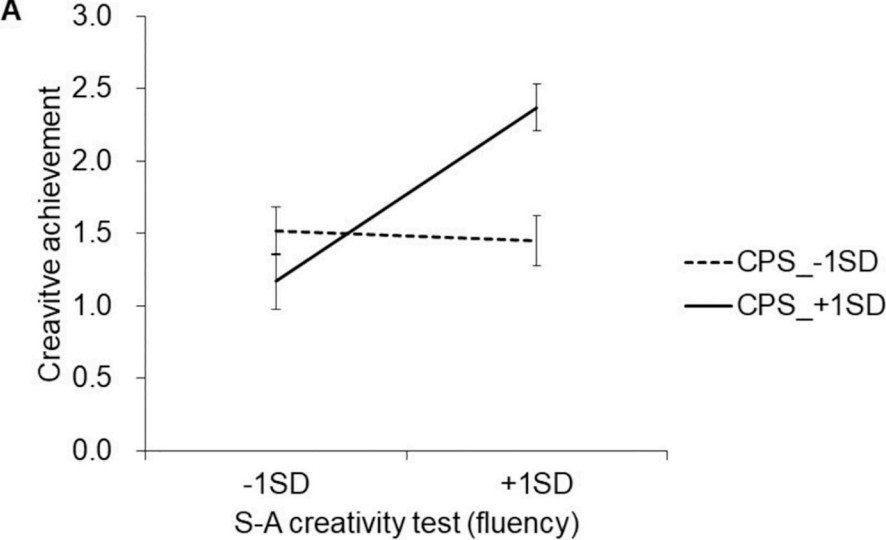

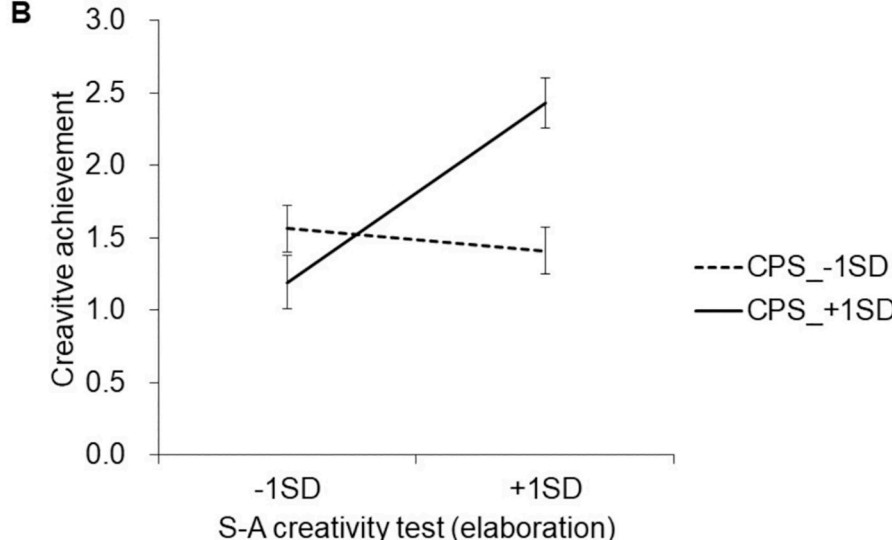

**Fig 5. Moderation effects of creative personality scale on the relationship between creative production score and creative achievement score.** (A) fluency, (B) elaboration. CPS: creative personality scale.

### Impact of beliefs about own creative personality on DT and creative behavior

Considering our results that beliefs about own creative personality can influence individuals' decisions regarding creative activities, such beliefs can be considered as key factors moderating the connection between DT and creative behaviors.

While the current study treated creative personality score as an index of beliefs about own creative personality—following the suggestions in the study by Kaufman [38]—other researchers have dealt with the creative personality score differently. For instance, Fürst and Grin used the score of openness/intellect as an index of creative personality [56]. According to the assumption by Lubart et al. [14], creative potential includes cognitive resources (e.g., DT) and

**Table 5. Results of multiple regression analysis in the effect of elaboration with creative production score as the objective variable.**

| Predictors | b | 95% CI | | T | df | p | β |
|---|---|---|---|---|---|---|---|
| | | LL | UL | | | | |
| Step 1 ($R^2$ = .094, p = .004) | | | | | | | |
| Intercept | 3.27 | 3.16 | 3.38 | 58.6 | 84 | < .0001 | |
| S-A Creativity test (elaboration) | 0.02 | 0.007 | 0.035 | 3.0 | 84 | .004 | .31 |
| Step 2 ($R^2$ = .127, p = .004) | | | | | | | |
| Intercept | 3.27 | 3.16 | 3.38 | 59.4 | 83 | < .0001 | |
| S-A Creativity test (elaboration) | 0.02 | 0.007 | 0.034 | 2.9 | 83 | .004 | .30 |
| CPS | 0.23 | -0.03 | 0.49 | 1.8 | 83 | .080 | .18 |
| Step 3 ($R^2$ = .139, p = .006) | | | | | | | |
| Intercept | 3.27 | 3.16 | 3.38 | 59.4 | 82 | < .0001 | |
| S-A Creativity test (elaboration) | 0.02 | 0.005 | 0.033 | 2.7 | 82 | .009 | .28 |
| CPS | 0.25 | -0.014 | 0.512 | 1.9 | 82 | .063 | .19 |
| S-A Creativity test (elaboration) × CPS | -0.02 | -0.052 | 0.016 | -1.1 | 82 | .294 | -.11 |

CI: confidential interval, LL: lower limits, UL: upper limits, CPS: creativity personality scale

conative resources (e.g., openness to experiences), and used DT and creative personality (openness) as psychological components. Then, these cited authors constructed a latent factor of creative potential by combining DT and openness, showing their impacts on creative production and achievements. This type of functional model of creative personality comprising DT and production and achievements is important when focusing on personalities, as personalities are considered to have close relationships with genetic and other innate factors (e.g., openness to experiences and novelty seeking) [57, 58]. Meanwhile, when measuring personality traits using self-assessed scales, the personality scores show individuals' own beliefs about each item. If we consider that the CPS measures beliefs about own creative personality, another function of conative resources of creative potential comes to light: Beliefs about own conative resources influence how DT is realized in actual creative behavior. Future research should examine the dual function of creative personality and beliefs about own creative personality in

**Table 6. Results of multiple regression analysis in the effect of elaboration with creative achievement score as the objective variable.**

| Predictors | b | 95% CI | | T | df | p | β |
|---|---|---|---|---|---|---|---|
| | | LL | UL | | | | |
| Step 1 ($R^2$ = .053, p = .031) | | | | | | | |
| Intercept | 1.66 | 1.47 | 1.84 | 17.8 | 85 | < .0001 | |
| S-A Creativity test (elaboration) | 0.03 | 0.002 | 0.049 | 2.2 | 85 | .031 | .23 |
| Step 2 ($R^2$ = .104, p = .010) | | | | | | | |
| Intercept | 1.66 | 1.48 | 1.84 | 18.2 | 84 | < .0001 | |
| S-A Creativity test (elaboration) | 0.03 | 0.002 | 0.048 | 2.2 | 84 | .031 | .23 |
| CPS | 0.47 | 0.042 | 0.91 | 2.2 | 84 | .032 | .23 |
| Step 3 ($R^2$ = .249, p < .0001) | | | | | | | |
| Intercept | 1.65 | 1.48 | 1.82 | 19.7 | 83 | < .0001 | |
| S-A Creativity test (elaboration) | 0.03 | 0.013 | 0.056 | 3.2 | 83 | .002 | .31 |
| CPS | 0.39 | -0.016 | 0.787 | 1.9 | 83 | .060 | .18 |
| S-A Creativity test (elaboration) × CPS | 0.10 | 0.053 | 0.157 | 4.0 | 83 | < .0001 | .39 |

CI: confidential interval, LL: lower limits, UL: upper limits, CPS: creativity personality scale

creativity actualization, as this may yield relevant data for better understanding the specific function of the Person dimension in the 4 Ps theory.

DT had a significantly weak association with creative production (fine arts). Additionally, it explained the increase in variance in creative production. This result is consistent with a previous study, which claimed that DT was positively related to creative output in poetry and storytelling in eighth-grade children (poetry: $r$ = .44, storytelling: $r$ = .45; Study 1 in Baer [21]). Since the current study could not use the data on creative production in the writing domain due to a lack of inter-rater reliability, we could not compare DT and creative production between the two domains used in the creative production tasks of the current paper. Furthermore, there are other creative domains (e.g., science, social activities, etc.) which have yet to be examined by scholars. Thus, future studies should explore the differences in the association between DT and creative production across different creative domains. Interestingly, the moderating role of beliefs about own creative personality was not consistent between creative behaviors (i.e., production and achievement). Instead, beliefs about own creative personality did not explain the increase in variance in creative production, nor was it significantly associated with creative production in fine arts. This low explanation from beliefs about own creative personality may be due to the settings of the creative production tasks proposed in the current study, which only dealt with an art domain. Accordingly, although the participants in this study had higher creative personality, they may not have been willing to utilize their DT in a cutouts task. In other words, beliefs about own creative personality regarding specific creative domains are likely to influence DT differently across tasks in different creative domains. Kaufman and Baer [40] showed a significant relationship between CPS and self-assessed creativity in specific creative domains, albeit the relationships were weak (e.g., $r$ = .23 in the art domain). Thus, future research should focus on beliefs about own creative personality in specific creative domains when making domain-specific examinations of the impact of such beliefs on the relationship between DT and creative production.

## Relationships among creativity indices

Researchers have developed a variety of methods to measure creativity, and accordingly examined the relationships among these methods when investigating the concurrent and predictive validity of the measures. Still, and despite decades of investigation [24–27, 31], the concurrent and predictive validity of DT tests remains controversial [59]. However, our findings have provided evidence for the concurrent validity of these tests. As in previous research that focused on DT and creative achievements [24, 25, 27, 28, 31], the current study only showed the positive association between DT and the total achievements score. This study used the CAQ, a popular measure of creative achievements. Nonetheless, the distribution of the results for the scores of this scale was often skewed depending on the nature of the target population and the data collection method. Since we collected data from a relatively small group (fewer than 100 people), we were able to use the overall scores in the analysis. Notwithstanding, other scales could be considered for assessing creative achievements when researchers see themselves not being able to use the scores for each domain. Specifically, there are various measures to assess creative achievements in general and specific domains, as follows: Creative Activity and Accomplishment Checklist [29]; Biographical Inventory of Creative Behaviors [60]; Kaufman Domains of Creativity Scale [61]; Inventory of Creative Activities and Achievements [62]. Hence, future studies should translate such measures into various languages and conduct creativity research with diverse cultural background groups. Further, although it is often difficult to recruit sufficient participants with higher creative achievements, a recent study proposed using a comparison group [63], as well as that researchers could control for variables of interest

such as age, gender, intelligence, and DT in the higher and lower creative achievement groups. Such a data collection method could be promising for future creativity research.

Although research on the relationship between DT and creative production is limited, Baer [21] (Study 1) showed a weak but positive relationship between fluency in verbal DT tasks and creative production tasks using a relatively small sample (eighth-grade children, N = 50). Baer [21] (Studies 2–5) also conducted similar experiments using small samples (sample range = 19–28) with different age groups (elementary school children and adults). The relationship between fluency in verbal DT tasks and creative production tasks was weakly positive, but it was not significant potentially due to the small sample groups. Therefore, Baer [22] replicated the experiments in another sample of eighth-grade children (N = 128) and showed small correlations between DT and creative production (story writing). The current study supports the findings in Baer's [21, 22] studies. Nonetheless, researchers could still further examine the correlation between DT and creative production across different creative domains.

## Conceptualization of creativity

Creativity is a very complex concept, so it comes as no surprise that numerous researchers tried to define it from various perspectives, imparted various categories to this concept, and have generally not integrated these categorizations and definitions. Based on this understanding, previous studies have proposed models of creativity in an attempt to unify its definitions and measurements [9, 10, 64], with one example being the 4 Ps theory [9].

The current study examined the associations of four creativity measures, each of which correspond to one of the 3 Ps: DT (Process), creativity production (Product), creativity achievement (Product), and beliefs about own creative personality (Person). The current findings revealed the associations between the variables representing the Process and Product dimensions and the moderating role of the variable representing the Person dimension (beliefs about own creative personality) in this relationship. The reason creative production and achievements were not correlated in the current study could be due to the misalignment in the variables that were compared, such as creative production in fine arts and total creative achievement scores.

Although the current study focused in three Ps of the 4 Ps theory, we must acknowledge the importance of Press in considering the relationships among creativity indices. It is evident that the Person and Process dimensions change depending on the environment. However, the creativity indices measured in this study included evaluations from others, such as the creative production and creative achievement. This evaluation process is influenced by domain, culture, and society, as suggested by the systems model in Csikszentmihalyi [6]. The systems model suggests that creative activities and products should be evaluated by experts in a specific domain and people belonging to a society and culture. The creative activities and products were situated within domains, a society, and culture through such processes. Future studies should endeavor to examine the influence of the dimension of Press on the dimensions of Process and Product, with one example being research on the relationship among various creativity factors integrating creativity conceptualization.

## Limitations and future studies

We should acknowledge several limitations of the current research. First, we attempted to measure DT using four subscales (i.e., Fluency, Flexibility, Originality, and Elaboration). Although it is standard to report on all DT subscales in related tasks [15, 23], originality and flexibility were not included in the current analyses due to inadequate reliability. Since these are two core concepts of DT, further research is necessary to examine whether these aspects

produce similar results. The low reliability of these subscales was caused by issues with the Japanese DT scoring process. The study scales were translated and redeveloped from the TTCT for Japan. However, we outsourced scoring to an external organization, entailing that the scoring process was closed. Hence, although tests like the TTCT, which have been translated worldwide, are convenient and easy to use, researchers should still pay attention to the validity and reliability of the scales and scoring processes in each test. Future studies in Japan should utilize a new scoring system—as described below—to ensure reliable and valid scoring.

The present study also examined creative production using the CAT method, which has been previously criticized regarding its generalizability [59]. Plucker et al. [59] implied that the validity of the rating process of the CAT (i.e., by experts in a given domain) has yet to be confirmed. As the current study failed to obtain an acceptable level of reliability for the Haiku rating, the CAT method for rating creative production cannot be generalized for other areas, rater types, cultures, and societies. Since Haiku is a unique Japanese literary expression, it is necessary to refine the product evaluation method in the future. Future studies should also examine the validity of the CAT method more in-depth.

Recently, researchers have indicated a procedural issue regarding creativity measurements in which scorers evaluate ideas or products. When scoring DT tasks or the CAT, the scorer should be trained in DT task scoring [65] or have sufficient knowledge and experience in a particular creative domain [15]. Studies have shown that despite recruiting such scorers, their scoring generally became burdensome, especially with large samples [65, 66]. To solve these issues, new scoring systems using Latent Semantic Analysis (LSA) [67] have been proposed in recent creativity research [68, 69]. LSA scoring has the potential to enable researchers to automatically score DT or literal products with high reliability and validity [65]. As such scoring systems have not yet been proposed in Japanese, future studies are necessary to develop these systems and conduct creativity research based on reliable and valid measurements in Japanese.

We also tried to examine creative production both generally and specifically; however, we could not measure the creative production score in multiple creative areas with an appropriate level of validity and reliability. Consequently, the present study could not ensure that DT consistently relates to creative production in multiple areas. Nevertheless, creativity measures are connected even if the creative domains are different.

Finally, although we tried to recruit the largest sample possible, we were not able to recruit more than 100 participants due to budget constraints and the potential burdens on participants and creative production task raters. Accordingly, although the current sample size (N = 88) has sufficient statistical power, it did not show sufficient stability for the estimates according to research by Schönbrodt and Perugini [70]; these authors posit that correlational studies should collect data from more than 150 participants for ensuring estimate stability. Thus, future researchers should try and replicate the current research with larger samples to confirm our findings.

## Supporting information

**S1 Table. Results of multiple regression analysis in the effect of flexibility with creative production score as the objective variable.**
(DOCX)

**S2 Table. Results of multiple regression analysis in the effect of originality with creative production score as the objective variable.**
(DOCX)

**S3 Table. Results of multiple regression analysis in the effect of flexibility with creative achievement score as the objective variable.**
(DOCX)

**S4 Table. Results of multiple regression analysis in the effect of originality with creative achievement score as the objective variable.**
(DOCX)

**S1 Dataset. Data used for the analysis reported in the article.**
(XLSX)

## Acknowledgments

We thank Koichi Amino, Haruna Funato, Akira Goto, Mitsuo Hayashi, Koji Nakada, and Sumio Suzuki for their help in conducting the study and Hiroyuki Noguchi for his advice on the analysis of this study. Editage provided English editing to improve the readability of the manuscript.

## Author Contributions

**Conceptualization:** Chiaki Ishiguro, Yuki Sato, Ai Takahashi, Yuko Abe, Etsuko Kato, Haruto Takagishi.

**Data curation:** Chiaki Ishiguro, Haruto Takagishi.

**Formal analysis:** Chiaki Ishiguro, Haruto Takagishi.

**Funding acquisition:** Yuki Sato.

**Methodology:** Yuki Sato, Ai Takahashi, Yuko Abe, Etsuko Kato.

**Project administration:** Yuki Sato, Haruto Takagishi.

**Supervision:** Etsuko Kato.

**Writing – original draft:** Chiaki Ishiguro.

**Writing – review & editing:** Yuki Sato, Ai Takahashi, Yuko Abe, Etsuko Kato, Haruto Takagishi.

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
