## [Decision Letter · Decision Letter 0]

2 Sep 2021

PONE-D-21-18303

The relationship between creativity indices: creative potential, production, achievement, and personality

PLOS ONE

Dear Dr. Takagishi,

Thank you for submitting your manuscript to PLOS ONE. After careful consideration, we feel that it has merit but does not fully meet PLOS ONE’s publication criteria as it currently stands. Therefore, we invite you to submit a revised version of the manuscript that addresses the points raised during the review process.

We have received the expert reviewers opinion on your submission. While it has some merits, reviewers agree that it needs to be improved further. Please look at all the comments and improvement recommendations brought forward in the reviews and act accordingly in producing a revised manuscript for further consideration by the panel of reviewers (the current as well as new reviewers will be invited).

We look forward to receiving your revised manuscript.

Kind regards,

Denis Alves Coelho, PhD

Academic Editor

PLOS ONE

“This research was supported by the Ministry of Education, Culture, Sports, Science and Technology (MEXT) as part of the Joint Research Program implemented at Tamagawa University Brain Science Institute in Japan.”

Reviewers' comments:

Reviewer's Responses to Questions

**Comments to the Author**

1. Is the manuscript technically sound, and do the data support the conclusions?

Reviewer #1: Partly

Reviewer #2: Yes

2. Has the statistical analysis been performed appropriately and rigorously? 

Reviewer #1: Yes

Reviewer #2: I Don't Know

3. Have the authors made all data underlying the findings in their manuscript fully available?

Reviewer #1: Yes

Reviewer #2: Yes

4. Is the manuscript presented in an intelligible fashion and written in standard English?

Reviewer #1: Yes

Reviewer #2: Yes

5. Review Comments to the Author

Reviewer #1: A strength of this paper is that it assumes that there's a difference between creative potential and creative productivity. This distinction is often overlooked, although there is a recent thread in the literature that recognizes the difference. The distinction is very important, so kudos to the authors.

Unfortunately, this manuscript has flaws. It is superficial. The authors did not do their homework. For example, in section 2.3 the authors failed to connect their measures of creative potential with existing measures. They do mention Guilford, but not in a way that really helps. They could have admitted that one of their tasks is the Uses Test, another is the Consequences Test, and another is Product Improvement. Many people besides Guilford have used these and mentioning that would help readers understand if there is any generalizability.

Even more problematic is that the current authors point to four indices, and some of these are not really a part of creativity theory. Rapidness and broadness, for example, are not theoretically tied to creativity. Rapidness sounds like something that is contrary to creativity. Gruber went into detail about how creative thinking takes time, and there are data from Mednick and other supporting that. Remote (original) associates take time and are not rapid. Beyond this, no operational definition of any of the indices is provided in this manuscript. This is a an enormous problem. Another problem is that a total score is formed from the four individual indices. That total score could be calculated in various ways, such as averaging across the four indices, or it could be an optimization of the four indices, but that is not described in the manuscript. There is debate in the research about totally vs using single indices, and usually the total score is rejected. All the authors do is refer to a external professional organization, one I have never heard of, even though I've been reading the creativity research for decades. You might say that there is much too much ambiguity here.

The authors would have been better off if they had a better sample of measures and tasks of creative potential. Right now they have three varied measures. It is much more common to use several items within any one type of test. It is more common for example, to have three Product improvement items, whereas the present authors only have one. Research often has three product improvement items and then three Uses items and then three Consequences items, or something like that. This allows authors to check the inter item reliability of their tests. Right now the authors have used only three items, and they are three items from three different kinds of tests, so you couldn't reasonably expect nor statistically check reliability. Again you could say that there is much too much ambiguity, here about reliability.

Ideally the authors would check reliability with their own sample because reliability always varies from sample to sample, and it is not satisfactory to simply report that they are using a measure that has demonstrated real reliability in the past with other samples. The Editor should watch for this in a revision. Do not allow them to simply refer to previous reliability.

Another concern is that the CAT method of judging products was used. And there are various problems discussed in the literature when using the cat. Various studies have shown that the results of CAT researech do not generalize: ratings from one group of judges are not well correlated with ratings from other groups of judges. Admittedly the CAT is often used but that doesn't mitigate the problem.

This research is not well tied to previous research. Indeed, the authors argue that the connection of potential to productivity has not really been studied, but as a matter of fact I know of several dozen studies which has explored that connection. It is a straw argument to say that it has not been studied. The present authors do cite a few studies, such as the longitudinal work using the Torrance data, but what of the very important seminal study of Wallach and Wing? And then Wallach did follow up studies, as did Nathan Kogan. Sure, these were done many years ago, but they arere quite meaningful and, again, it is incorrect to say that the connection has not been studied. results, period. I would also suggests that they look at the work of Roberta Milgram on potential as connected to actual creative productivity and performance, and Plucker’s re-analysis of the Torrance data, and Runco’s 2-3 studies of creative potential connected to creative behavior. It is poor scholarship to overlook so much relevant research.

Then there is the fact that the Creative Achievement Questionnaire has also been criticized in a convincing fashion. Results from this measure tend to be skewed, and the theoretical basis for the questions within the CAT have been criticized a number of times. Those questions were nominated by a small group of judges and are therefore limited and potentially biased. There is a much better measure called the Creative Activity and Accomplishment Checklist (CAAC), which was developed a long ago by Holland, and then refined in a dozen or more studies since then. Statistically and theoretically it is a much present better measure than the CAQ. An and then Paek have recently published using the newer, better CAAC. Why do I mention this? I know the present authors will not want to go back and collect new data with the better measure, but their choice and their ignoring the problems with the CAQ is another indication that they did not really do their homework. The authors look a bit foolish when they claim on line 47 “that most of the studies measuring creative behavior use the Creative Achievement questionnaire or the Biographical Inventory." The CAAC has been used 3-4 times as often as the CAQ, no doubt because it is a better measure. AND the BIC is really just a copy of the CAAC.

The authors claim that the link between creative potential and behavior is less than .3. That is often true, but there is research showing the connection using measures of creative potential, that are similar to those used here in with results of up to .55. Runco (1986) used mult regression with tests similar to the ones used herein and the criterion measure mentioned above and found that .55. Here again I don’t think the authors did a good job looking at previous research. One more example: Plucker did include a personality trait in his study of divergent thinking (creative potential) and creative behavior. He made the important point that tests of potential are often not aligned with the criteria used in the research. That would include the CAQ. Plucker et al. used a criterion that was aligned with the predictors–a criterion based on ideation (which is what the predictors measure).

I did not see probability level levels or degrees of freedom in the regression table. I am accustomed to APA tables, which require that all information is provided within each table; that the table is self-contained, self-sufficient. The authors report probabilities in the body of the paper but in the Tables. Degrees of freedom seem to be missing here and there (e.g., line 214).

I would not put divergent thinking in the Person category of the 4Ps. It is a process measure (how ideas are associated with one another) OR a measure of products (ideas = products). The Person category = traits and the like. DT is not a trait. It reflects a capacity to produce ideas, often original. This is a small point, I know, but if the ms is revised, I would change this. (And there is a 6P model, by the way, called the hierarchical model, but that too is a small point.)

Reviewer #2: The relationship between creativity and cognition forms a central domain of the current scientific development in the description and explanation of human behaviour and its creative process, thus the study “The relationship between creativity indices: creative potential, production, achievement, and personality” fits very well in current creativity research. The paper is well written and thoroughly researched.

The methodological approach could be better explained. For someone who doesn’t have knowledge about descriptive statistics and measurement variables, the graphics are difficult to understand.

It would also be nice to have some pictures of the “creative productions”: for example, a haiku poem or a cut out paper object. \\n

The following formal structure would make the paper clearer:

1. Introduction

1.1. Approaches to the Creativity Concept

1.2. Ways of Measuring Creativity Indices

1.3. Research Objectives \\n

2. Methodological Approach \\n

2.1. Participants

2.2. Procedure \\n

2.3. Creative Potential Test

2.4. Creative Production Task

2.5. Rating Product Creativity \\n

3. Results \\n

4. Discussion

4.1. Impact of Personality on Creative Potential and Behaviour \\n

4.2. Relationships between Creativity Indices \\n

4.3. Conceptualisation of Creativity \\n

5. Limitations and Future Studies \\n

My last observation is related to the approach to Creativity. In this paper elements such as the social or cultural background of a creative individual in relation to ‘creative potential’ and ‘behaviour' are not considered. The authors of the study could include the systemic perspective of creativity which considers several factors, even including the role of the specialists who evaluate the creative achievements. I recommend reading articles or books from Csikszentmihalyi about the systemic perspective of Creativity.

6. PLOS authors have the option to publish the peer review history of their article (what does this mean?). If published, this will include your full peer review and any attached files.

Reviewer #1: No

Reviewer #2: **Yes: **Katja Tschimmel

---

## [Author Response · Author response to Decision Letter 0]

10 Mar 2022

I apologize for the delay in revising the manuscript. We thank that both reviewers provided us constructive comments and suggestions. Based on reviewers’ fruitful suggestions, we amended our manuscript thoroughly. We would appreciate if you find below our responses to each comments. 

I found one mistake regarding an old manuscript. The old manuscript incorrectly handled zeros when converting creativity achievement values to logarithmic values. In the new manuscript, I did a logarithmic conversion using all values plus one. The results section has been amended accordingly. The results remain unchanged from the old manuscript.

Reviewer #1:

Comment 1. Unfortunately, this manuscript has flaws. It is superficial. The authors did not do their homework. For example, in section 2.3 the authors failed to connect their measures of creative potential with existing measures. They do mention Guilford, but not in a way that really helps. They could have admitted that one of their tasks is the Uses Test, another is the Consequences Test, and another is Product Improvement. Many people besides Guilford have used these and mentioning that would help readers understand if there is any generalizability.

Response 1. The previous manuscript missed to explain the specific relationship between S-A creativity test and the previous divergent thinking tests. S-A creativity test was developed to assess DT under the supervision of Guilford and with reference to TTCT and the high reliability and validity were reported in previous research for Japanese participants. The subtests and the scoring process (in the four aspects: Fluency, Flexibility, Originality and Elaboration) are based on TTCT. The current manuscript illustrates this information with more precise reference to previous tests and research. 

p. 8, l. 163: Creative potential was measured using the S-A Creativity test [42], standardized for Japanese participants. J.P. Guilford supervised the development of this instrument to assess DT based on the TTCT. It has high reliability and validity and can be administered to 4th grade to university-level students [42]. S-A Creativity test scores have been applied to various research studies related to creativity in Japan. Specifically, the test showed the positive associations between personality and problem-solving in daily life [43]. It also showed the relationship between neurological and physiological data (e.g., [44,45]).

The test is composed of three types of subtests: Unusual Uses Task, Product Improvement, and Just Suppose in TTCT. The Unusual Uses Task subtest requires participants to generate unique ways of using typical objects, such as a newspaper, button, and chopsticks. For example, participants were asked the following question: “Other than reading, what other uses are there for newspapers? For instance, you may use them to wrap things.” The Product Improvement subtest requires participants to imagine the different ways a product, such as a TV, pot, or desk, can be improved. For instance, participants were asked, “What kind of TV can you imagine having? For example, a TV that can show 3D images. Write down as many as possible.” Finally, the Just Suppose subtest requires participants to imagine the consequences when something unimaginable happens, such as when there are no clocks in the world. For example, participants were asked, “What would happen if all the mice in the world disappeared? One possible consequence is that cats might become hungry.” Participants are given two minutes for a trial and five minutes for two questions in each subtest. 

Comment 2. Even more problematic is that the current authors point to four indices, and some of these are not really a part of creativity theory. Rapidness and broadness, for example, are not theoretically tied to creativity. Rapidness sounds like something that is contrary to creativity. Gruber went into detail about how creative thinking takes time, and there are data from Mednick and other supporting that. Remote (original) associates take time and are not rapid. Beyond this, no operational definition of any of the indices is provided in this manuscript. This is a an enormous problem. Another problem is that a total score is formed from the four individual indices. That total score could be calculated in various ways, such as averaging across the four indices, or it could be an optimization of the four indices, but that is not described in the manuscript. There is debate in the research about totally vs using single indices, and usually the total score is rejected. All the authors do is refer to a external professional organization, one I have never heard of, even though I've been reading the creativity research for decades. You might say that there is much too much ambiguity here.

Response 2. The previous manuscript inappropriately translated the four aspects (Fluency, Flexibility, Originality and Elaboration) into (Rapidness, Broadness, Originality and Elaboration). because we retranslated the relatively classic Japanese translation of the four aspects in 50 years ago to English without considering the correspondence to TTCT. Precisely, the S-A creativity test was developed with referencing the TTCT and the aspects should have been translated into Fluency, Flexibility, Originality and Elaboration. The present manuscript illustrated the correspondence of S-A creativity test and TTCT and appropriately translated the four aspects based on each definition and scoring procedure. 

Although S-A creativity test generally use the sum of Originality and Elaboration score because of the higher correlation of Fluency with Flexibility and Elaboration, TTCT discouraged to use the total score of the aspects. Further, the reliabilities of each scoring should be clarified in each sample. Therefore, we asked the professional organization to show the raw scores of the test in the current sample and calculated the reliabilities and correlations. As a result, the reliabilities of Flexibility and Originality were less than .70, which may distort the estimation of true correlations. In addition, the correlation of Fluency and Elaboration was quite high (r = .71), which is difficult to discriminate the two aspects. Therefore, the current study decided to use the only Fluency for the further analyses. 

p. 9, l. 170: The test is composed of three types of subtests: Unusual Uses Task, Product Improvement, and Just Suppose in TTCT. The Unusual Uses Task subtest requires participants to generate unique ways of using typical objects, such as a newspaper, button, and chopsticks. For example, participants were asked the following question: “Other than reading, what other uses are there for newspapers? For instance, you may use them to wrap things.” The Product Improvement subtest requires participants to imagine the different ways a product, such as a TV, pot, or desk, can be improved. For instance, participants were asked, “What kind of TV can you imagine having? For example, a TV that can show 3D images. Write down as many as possible.” Finally, the Just Suppose subtest requires participants to imagine the consequences when something unimaginable happens, such as when there are no clocks in the world. For example, participants were asked, “What would happen if all the mice in the world disappeared? One possible consequence is that cats might become hungry.” Participants are given two minutes for a trial and five minutes for two questions in each subtest.

The test is scored by trained judges on four aspects: Fluency, Flexibility, Originality, and Elaboration. Fluency is a measure of the ability to generate many ideas. It is evaluated by the number of responses, excluding those that are inappropriate or difficult to interpret. Flexibility is the ability to generate ideas from a wide range of perspectives. It is assessed by the number of categories in which ideas are generated based on the criterion table or equivalent judgment. The criterion table indicates a particular answer’s classification and how many points it receives. Originality is the ability to generate ideas different from those of others. It is evaluated based on the total number of categories weighted by the criterion table or an almost equivalent judgment; the criterion table indicates how many points should be given based on how rare the responses are in each category. Finally, Elaboration is the ability to think concretely about ideas. It is evaluated based on the total number of responses weighted by the criterion table or equivalent judgments. Torrance [46] initially established these four aspects as perspectives for evaluating ideas based on the elements of DT proposed by Guilford [9]. An external professional organization evaluated all of the ratings (Tokyo Shinri Corporations) [42].

p. 13, l. 258: The scores for Fluency and Flexibility highly correlated with those for Elaboration in the past surveys for Japanese. Therefore, the total creativity score was defined as the sum of the raw score of Originality and Elaboration in the S-A Creativity test. However, Guilford [9] viewed DT as multidimensional, and Torrance [46] discouraged researchers from using the total score of the four aspects. Since the use of the total score of the TTCT is controversial (e.g., [31,49]), the current study first checked the reliability of scoring of each aspect in the current sample, then examined the relationships among the aspects. Although the reliability score is required to be more than .80, or at least more than .70 in psychometric assessment [50], the current results showed that the reliability scores (Cronbach alpha) of some evaluation aspects were not acceptable (Fluency: .80; Flexibility: .67; Originality: .56; Elaboration: .77). On the other hand, the reliabilities of Fluency and Elaboration were adequate. Still, the correlation of the two aspects was relatively high (r(86) = .889, p < .0001). Therefore, the current study used the score for Fluency as the most reliable rating with similar characteristics to the other perspectives in subsequent analyses.

Comment 3. The authors would have been better off if they had a better sample of measures and tasks of creative potential. Right now they have three varied measures. It is much more common to use several items within any one type of test. It is more common for example, to have three Product improvement items, whereas the present authors only have one. Research often has three product improvement items and then three Uses items and then three Consequences items, or something like that. This allows authors to check the inter item reliability of their tests. Right now the authors have used only three items, and they are three items from three different kinds of tests, so you couldn't reasonably expect nor statistically check reliability. Again you could say that there is much too much ambiguity, here about reliability.

Ideally the authors would check reliability with their own sample because reliability always varies from sample to sample, and it is not satisfactory to simply report that they are using a measure that has demonstrated real reliability in the past with other samples. The Editor should watch for this in a revision. Do not allow them to simply refer to previous reliability.

Response 3. The previous manuscript could not explain the specific procedure and scoring process in an appropriate way. The S-A Creativity test included three subtests that correspond to Unusual Uses Test, Product Improvement and Just Suppose. Participants are provided two minutes for a trial and five minutes for two questions in each subtest, which were scored in four aspects. As mentioned above, we also calculated the reliabilities of scoring in the four aspects. With considering the reliabilities of each scoring and the relationships between the four scores, the current study decided to use only Fluency for the subsequent analyses. These information were demonstrated in the procedure and results sections.

p. 9, l. 170: The test is composed of three types of subtests: Unusual Uses Task, Product Improvement, and Just Suppose in TTCT. The Unusual Uses Task subtest requires participants to generate unique ways of using typical objects, such as a newspaper, button, and chopsticks. For example, participants were asked the following question: “Other than reading, what other uses are there for newspapers? For instance, you may use them to wrap things.” The Product Improvement subtest requires participants to imagine the different ways a product, such as a TV, pot, or desk, can be improved. For instance, participants were asked, “What kind of TV can you imagine having? For example, a TV that can show 3D images. Write down as many as possible.” Finally, the Just Suppose subtest requires participants to imagine the consequences when something unimaginable happens, such as when there are no clocks in the world. For example, participants were asked, “What would happen if all the mice in the world disappeared? One possible consequence is that cats might become hungry.” Participants are given two minutes for a trial and five minutes for two questions in each subtest.

The test is scored by trained judges on four aspects: Fluency, Flexibility, Originality, and Elaboration. Fluency is a measure of the ability to generate many ideas. It is evaluated by the number of responses, excluding those that are inappropriate or difficult to interpret. Flexibility is the ability to generate ideas from a wide range of perspectives. It is assessed by the number of categories in which ideas are generated based on the criterion table or equivalent judgment. The criterion table indicates a particular answer’s classification and how many points it receives. Originality is the ability to generate ideas different from those of others. It is evaluated based on the total number of categories weighted by the criterion table or an almost equivalent judgment; the criterion table indicates how many points should be given based on how rare the responses are in each category. Finally, Elaboration is the ability to think concretely about ideas. It is evaluated based on the total number of responses weighted by the criterion table or equivalent judgments. Torrance [46] initially established these four aspects as perspectives for evaluating ideas based on the elements of DT proposed by Guilford [9]. An external professional organization evaluated all of the ratings (Tokyo Shinri Corporations) [42].

p. 13, l. 258: The scores for Fluency and Flexibility highly correlated with those for Elaboration in the past surveys for Japanese. Therefore, the total creativity score was defined as the sum of the raw score of Originality and Elaboration in the S-A Creativity test. However, Guilford [9] viewed DT as multidimensional, and Torrance [46] discouraged researchers from using the total score of the four aspects. Since the use of the total score of the TTCT is controversial (e.g., [31,49]), the current study first checked the reliability of scoring of each aspect in the current sample, then examined the relationships among the aspects. Although the reliability score is required to be more than .80, or at least more than .70 in psychometric assessment [50], the current results showed that the reliability scores (Cronbach alpha) of some evaluation aspects were not acceptable (Fluency: .80; Flexibility: .67; Originality: .56; Elaboration: .77). On the other hand, the reliabilities of Fluency and Elaboration were adequate. Still, the correlation of the two aspects was relatively high (r(86) = .889, p < .0001). Therefore, the current study used the score for Fluency as the most reliable rating with similar characteristics to the other perspectives in subsequent analyses.

Comment 4. Another concern is that the CAT method of judging products was used. And there are various problems discussed in the literature when using the cat. Various studies have shown that the results of CAT researech do not generalize: ratings from one group of judges are not well correlated with ratings from other groups of judges. Admittedly the CAT is often used but that doesn't mitigate the problem.

Response 4. The previous manuscript could not specify the generalizability of the CAT results. The current study measured creative potential and production in a certain group with using the method of CAT. However, as you mentioned, the scoring results could be different if the other experts rated the product creativity. The present manuscript discussed this point as a limitation of generalizability of the current study in the discussion section.

p. 22, l. 433: The present study also examined creative production using the CAT method, criticized for its generalizability [35]. Plucker et al. [34] implied that the CAT includes a rating process by experts in the domain, and the validity of the rating process is still under consideration. As the current study failed to obtain an acceptable level of reliability for the Haiku rating, the CAT method for rating creative production cannot be generalized according to each area, rater type, culture, and society. Since Haiku is a unique Japanese literary expression, it is necessary to refine the product evaluation method in the future. Future studies should also examine the validity of the CAT method more specifically.

Comment 5. This research is not well tied to previous research. Indeed, the authors argue that the connection of potential to productivity has not really been studied, but as a matter of fact I know of several dozen studies which has explored that connection. It is a straw argument to say that it has not been studied. The present authors do cite a few studies, such as the longitudinal work using the Torrance data, but what of the very important seminal study of Wallach and Wing? And then Wallach did follow up studies, as did Nathan Kogan. Sure, these were done many years ago, but they arere quite meaningful and, again, it is incorrect to say that the connection has not been studied. results, period. I would also suggests that they look at the work of Roberta Milgram on potential as connected to actual creative productivity and performance, and Plucker’s re-analysis of the Torrance data, and Runco’s 2-3 studies of creative potential connected to creative behavior. It is poor scholarship to overlook so much relevant research.

Response 5. The previous manuscript could not review the previous research on creativity. We appreciated your suggestion of previous articles and reviewed further research as long as we can access. Then, we reviewed the previous findings and reconsidered the findings of the current study. We believe that the current study has implications to illustrate the specific relationship of creative potential, production, achievements and personality. In addition, these relationships in a young adult group are suggestive to understand the interdisciplinary studies on creativity because such studies separately examine the relationships between a variety of creativity indices and biological measures in each study. The detailed review and the significance of the current study were specified in the introduction section.

p. 3, l. 54: Psychologists have measured such test scores to assess individuals’ creative potential, which is expected to relate to and predict their creative production and performance due to the Product factor of the 4 Ps theory [2]. However, this association between creative potential and performance is weakly positive in most studies conducted on children to adults (e.g., [13–17]). The predictive role of creative potential has also been examined but remains controversial. Kogan and Panvoke [13] showed that children’s creative potentials correlated with their accomplishments in the 5th but not in the 10th grade. The 18-year follow-up study also showed that the creative potential of high school students was only associated with specific areas of creative achievements [18]. Torrance [10] designed the Torrance Test of Creative Thinking (TTCT) to indicate creative potential that promotes creative behaviors. In addition, previous empirical research with a 40- or 50-year follow-up found that creative ideation scores predict future creative achievement [19–21]. Further, it is suggested in a meta-analysis [22] and follow-up study [19] that the TTCT has a better predictive association of creative achievements than IQ. However, it has been argued that the link between creative potential and behavior is weak (r score is less than .30) and sometimes insignificant [23]. An examination of what causes this weak correlation follows

.….

In recent years, the number of interdisciplinary studies on the neurological and genetic basis of creativity (e.g., [38–41]) has increased significantly. However, each one employed a different creativity index. Therefore, clarifying the relationship between creativity and the rest of the human biological substrate, and that among creativity indices, will help in explaining the biological basis of creativity.

Comment 6. Then there is the fact that the Creative Achievement Questionnaire has also been criticized in a convincing fashion. Results from this measure tend to be skewed, and the theoretical basis for the questions within the CAT have been criticized a number of times. Those questions were nominated by a small group of judges and are therefore limited and potentially biased. There is a much better measure called the Creative Activity and Accomplishment Checklist (CAAC), which was developed a long ago by Holland, and then refined in a dozen or more studies since then. Statistically and theoretically it is a much present better measure than the CAQ. An and then Paek have recently published using the newer, better CAAC. Why do I mention this? I know the present authors will not want to go back and collect new data with the better measure, but their choice and their ignoring the problems with the CAQ is another indication that they did not really do their homework. The authors look a bit foolish when they claim on line 47 “that most of the studies measuring creative behavior use the Creative Achievement questionnaire or the Biographical Inventory." The CAAC has been used 3-4 times as often as the CAQ, no doubt because it is a better measure. AND the BIC is really just a copy of the CAAC.

Response 6. The previous manuscript did not mention the CAAC. As you mentioned, however, there are a variety of measurements of creative achievements other than CAQ. The current study chose CAQ because it was one of the popular measurements of creative achievement and because it was composed of questions that is applicable for Japanese participants. Still, the score of CAQ is often skewed (actually it was skewed in the current study) as you mentioned. The present manuscript warned this problem as the limitation of the current study and suggested that researchers should consider to select the other measurements or to consider the way to collect the appropriate participants in each research objective.

p. 19, l. 366: As in previous research that focused on creative potential and achievements (e.g., [13,14,16,17,19]), the current study only showed the positive association between creative potential and the total achievements score. This study used the CAQ, a popular measure of creative achievements. However, the distribution is often skewed depending on the nature of the target population and the data collection method. Since we collected study data from a relatively small group (fewer than 100 people), the overall scores could be used in the analysis. However, another measure for creative achievements should be considered when addressing the problem of not using the scores for each domain. As there are various measures to assess creative achievements in general and specific domains (e.g., Creative Activity and Accomplishment Checklist [24]; Runco Ideational Behavior Scale [53]; Biographical Inventory of Creative Behaviors [26]; Kaufman Domains of Creativity Scale [54]; Inventory of Creative Activities and Achievements [55]), future studies should translate such measures into various languages and conduct creativity research for diverse cultural background groups. Further, although it is often difficult to gather sufficient participants with higher creative achievements, a recent study proposed using a comparison group. Researchers could control variables of interest such as age, gender, intelligence, and creative potential in the higher and lower creative achievement groups [56]. Such a data collection method could be promising for future creativity research.

Comment 7. The authors claim that the link between creative potential and behavior is less than .3. That is often true, but there is research showing the connection using measures of creative potential, that are similar to those used here in with results of up to .55. Runco (1986) used mult regression with tests similar to the ones used herein and the criterion measure mentioned above and found that .55. Here again I don’t think the authors did a good job looking at previous research. One more example: Plucker did include a personality trait in his study of divergent thinking (creative potential) and creative behavior. He made the important point that tests of potential are often not aligned with the criteria used in the research. That would include the CAQ. Plucker et al. used a criterion that was aligned with the predictors–a criterion based on ideation (which is what the predictors measure).

Response 7. The previous manuscript lacked enough explanation of the statistics. As you mentioned, Runco (1986) showed the moderate correlation with creative potential and achievement in a specific domain within the gifted children. The current study was conducted an experiment for young adults because the finding could be suggestive for understanding the recent interdisciplinary research on creativity. In the amendment manuscript, seminal multi-regression analyses for creativity indices with referencing the Plucker et al (2010) as you suggested. 

p. 14, l. 280: Next, we examined the moderating effect of the CPS score on the association between creative potential and creative behavior. First, a multiple regression analysis was conducted with the creative production of art as the objective variable and creative potential score, CPQ score, and the interaction term between creative potential score and CPQ score as explanatory variables (Table 2). In Step 1, we examined the main effects of creative potential score on creative products and found that creative potential score had a positive effect (β = .26, t(84) = 2.4, p = .018). In Step 2, the CPS score was added to the explanatory variables, the effect of the fluency score remained (β = .24, t(83) = 2.3, p = .027) but there was no effect of the CPS (β = .16, t(83) = 1.5, p = .125). In Step 3, the interaction term between the creative potential score and the CPS score was added to the explanatory variables. Still, there was no interaction effect between the creative potential score and the CPS score (β = -.17, t(82) = -1.6, p = .117).

In addition, a multiple regression analysis was conducted with the CAQ score as the objective variable, creative potential and CPS scores as the explanatory variables, and the interaction term between the creative potential score and the CPS score (Table 3). In Step 1, we examined the main effects of the creative potential score on the CAQ score and found that the creative potential score had a positive effect (β = .31, t(85) = 3.0, p = .004). Next, the results of the analysis in Step 2, in which the CPS score were added to the explanatory variables, showed a positive effect of the creative potential score (β = .29, t(84) = 2.8, p = .006), but no in the CPS score (β = .20, t(84) = 2.0, p = .053). In Step 3, the interaction terms of the creative potential score and the CPS score were added to the explanatory variables, showed an interaction effect between the creative potential score and the CPS score (β = .33, t(83) = 3.5, p = .001, ΔR2 = .110). The simple main effect test results showed that the creative potential score had a positive effect on the CAQ score in the high CPS group (β = .68, t(83) = 4.6, p < .0001, Figure 5). However, the creative potential score had no effect in the high CPS group (β = -.04, t(83) = 0.3, p = .767).

Comment 8. I did not see probability level levels or degrees of freedom in the regression table. I am accustomed to APA tables, which require that all information is provided within each table; that the table is self-contained, self-sufficient. The authors report probabilities in the body of the paper but in the Tables. Degrees of freedom seem to be missing here and there (e.g., line 214).

Response 8. The previous paper did not include information on degrees of freedom and probability. In this manuscript, we have added a table that covers information about these, and the results of the regression analysis are described in the text, including the degrees of freedom.

see Table 2 Table 3 and Results part: 

Comment 9. I would not put divergent thinking in the Person category of the 4Ps. It is a process measure (how ideas are associated with one another) OR a measure of products (ideas = products). The Person category = traits and the like. DT is not a trait. It reflects a capacity to produce ideas, often original. This is a small point, I know, but if the ms is revised, I would change this. (And there is a 6P model, by the way, called the hierarchical model, but that too is a small point.)

Response 9. After reviewing the contents of the 4P’s theory, as you pointed out, DTs are positioned to measure the cognitive process during idea generation. Therefore, in the revised paper, we discussed the relationship between creative potential as Process, production and achievements as Product, and personality as Person.

p. 20, l. 406: For instance, creativity can be distinguished depending on whether it is attributable to the Person, Process, Press (environment), and Product (4 P’s theory) [7]. This 4 P model has been further developed into the 6 P model [61], adding Purpose and Problems. The current study examined the associations of four creativity measures: creative potential, production, achievement, and personality, which correspond to Person (personality), Process (creative potential), and Product (production and achievements). The current findings revealed the associations between Process and Product and the moderating role of personality in this relationship. The reason creative production and achievements were not correlated in the current study could be due to the misalignment in comparison areas, such as production art and total score for achievement.

 

Reviewer #2:

Comment 1. The methodological approach could be better explained. For someone who doesn’t have knowledge about descriptive statistics and measurement variables, the graphics are difficult to understand.

Response 1. The previous manuscript did not fully explain the variables measured and the statistical explanations. In the revised manuscript, I provided more detailed and clear explanation.

p. 11, l. 234: A series of self-report questionnaires measured participants’ creative achievement and creative personality. The first questionnaire was the CAQ by Carson et al. [25], which measured achievement in the 10 creative domains (fine arts, music, dance, architectural design, writing, comedy, invention, science, theater and film, and cooking). Total achievement scores in all domains were used in the analysis. The second was the CPS [37], which consists of 20 items selected from the International Personality Item Pool [47,48], which is a 5-point Likert scale (1: not at all to 5: very much) and measures creativity in general and in specific domains (e.g., science, interpersonal communication, writing, art). In this study, the overall mean was used for the analysis.

p. 12, l. 244: The descriptive statistics of the S-A Creativity test, creative production (in fine arts), total score of the CAQ, and the CPS score are shown in Table 1. The distribution of scores for each task is illustrated in Figure 3. A Shapiro-Wilk test was used to test whether scores on each measure of creativity were normally distributed, including fluency (W = 0.97, p = .053), flexibility (W = 0.98, p = .102) and originality (W = 0.98, p = .402), elaboration (W = 0.99, p = .813), creative production (W = 0.98, p = .180) and CPS score (W = 0.98, p = .196) were normally distributed, but the creative achievement score was not (W = 0.77, p < 0.0001). Therefore, log-transformed values of the creative achievement scores were used in the analysis.

Comment 2. It would also be nice to have some pictures of the “creative productions”: for example, a haiku poem or a cut out paper object. 

Response 2. In the previous manuscript, there were no images of paper cutouts or poems, which may have made it difficult to get a concrete image. In this paper, I have included images of paper cutouts as Figure 2 for each theme.

See Figure 2

Comment 3. The following formal structure would make the paper clearer:

1. Introduction

1.1. Approaches to the Creativity Concept

1.2. Ways of Measuring Creativity Indices

1.3. Research Objectives \\n

2. Methodological Approach \\n

2.1. Participants

2.2. Procedure 

2.3. Creative Potential Test

2.4. Creative Production Task

2.5. Rating Product Creativity 

3. Results 

4. Discussion

4.1. Impact of Personality on Creative Potential and Behaviour \\n

4.2. Relationships between Creativity Indices \\n

4.3. Conceptualisation of Creativity \\n

5. Limitations and Future Studies 

Response 3. The structure of the paper was not clear, so we followed the format you gave me and considered the structure. As you pointed out, it was necessary to describe the research approach to creativity in the introduction. Due to the flow of the paper, I could not add a section, but I added the contents related to the approach to the introduction.

p. 3, l. 40: Creativity is a uniquely human ability that psychological researchers have claimed is demonstrated by each individual in their everyday lives [1,2]. However, because it has a variety of attributes due to conceptual vagueness, researchers have classified the types of creativity according to various approaches ranging from cognitive and personality approaches to systems and sociocultural theories (e.g., [3–7]). For instance, the 4 Ps theory claims that creativity could be attributed to the Person (individuals’ cognitive and metacognitive traits), Process (psychological process to perform creative acts), Press (environment and situations that produce creativity), and Product (specific works and achievement to demonstrate creativity) [8]. While researchers have developed various instruments and methods to measure creativity based on such theoretical frameworks, the relationships among creativity indices remain unclear.

Comment 4. My last observation is related to the approach to Creativity. In this paper elements such as the social or cultural background of a creative individual in relation to ‘creative potential’ and ‘behaviour' are not considered. The authors of the study could include the systemic perspective of creativity which considers several factors, even including the role of the specialists who evaluate the creative achievements. I recommend reading articles or books from Csikszentmihalyi about the systemic perspective of Creativity.

Response 4. As you pointed out, socio-cultural background is an important factor in creativity research. In fact, creative achievements are especially those that are produced by each research collaborator as they are influenced by the domain, society, and culture. This is a part of the press of the 4P's theory, but we could not examine this influence in this study. Therefore, in this paper, I have explained this issue as a limitation of the research, citing the Systems Model.

p. 21, l. 415: Although the current study focused on three of the 4 P models, Press is also essential in considering the relationships among creativity indices. It is evident that Person and Process change depending on the environment. Still, indices, such as creative production and creative achievement, measured in this study, include evaluations from others. This evaluation process is influenced by domain, culture, and society, as suggested by the systems model in Csikszentmihalyi [4]. The systems model suggested that creative activity and products were evaluated by experts in a specific domain and people belonging to a society and culture. The creative activities and products were situated within domains, a society, and culture through such processes. Future studies should illustrate such an influence by Press on Process and Product. For example, we could research the relationship among various creativity factors integrating creativity conceptualization.

---

## [Decision Letter · Decision Letter 1]

7 Apr 2022

PONE-D-21-18303R1Relationships among creativity indices: Creative potential, production, achievement, and personalityPLOS ONE

Dear Dr. Takagishi,

Thank you for submitting your manuscript to PLOS ONE. After careful consideration, we feel that it has merit but does not fully meet PLOS ONE’s publication criteria as it currently stands. Therefore, we invite you to submit a revised version of the manuscript that addresses the points raised during the review process. Please submit your revised manuscript by May 22 2022 11:59PM. If you will need more time than this to complete your revisions, please reply to this message or contact the journal office at plosone@plos.org. Please include the following items when submitting your revised manuscript:A rebuttal letter that responds to each point raised by the academic editor and reviewer(s). You should upload this letter as a separate file labeled 'Response to Reviewers'.A marked-up copy of your manuscript that highlights changes made to the original version. You should upload this as a separate file labeled 'Revised Manuscript with Track Changes'.An unmarked version of your revised paper without tracked changes. You should upload this as a separate file labeled 'Manuscript'.If applicable, we recommend that you deposit your laboratory protocols in protocols.io to enhance the reproducibility of your results. Protocols.io assigns your protocol its own identifier (DOI) so that it can be cited independently in the future. For instructions see: https://journals.plos.org/plosone/s/submission-guidelines#loc-laboratory-protocols. Additionally, PLOS ONE offers an option for publishing peer-reviewed Lab Protocol articles, which describe protocols hosted on protocols.io. Read more information on sharing protocols at https://plos.org/protocols?utm_medium=editorial-email&utm_source=authorletters&utm_campaign=protocols.

We look forward to receiving your revised manuscript.

Kind regards,

Denis Alves Coelho, PhD

Academic Editor

PLOS ONE

Journal Requirements:

Additional Editor Comments (if provided):

Reviewer 2 has noticed some need of further clarification and based on this advice we are considering your manuscript as accepted pending minor revision.

Reviewers' comments:

Reviewer's Responses to Questions

**Comments to the Author**

1. If the authors have adequately addressed your comments raised in a previous round of review and you feel that this manuscript is now acceptable for publication, you may indicate that here to bypass the “Comments to the Author” section, enter your conflict of interest statement in the “Confidential to Editor” section, and submit your "Accept" recommendation.

Reviewer #1: All comments have been addressed

Reviewer #2: All comments have been addressed

2. Is the manuscript technically sound, and do the data support the conclusions?

Reviewer #1: Partly

Reviewer #2: Yes

3. Has the statistical analysis been performed appropriately and rigorously? 

Reviewer #1: Yes

Reviewer #2: I Don't Know

4. Have the authors made all data underlying the findings in their manuscript fully available?

Reviewer #1: Yes

Reviewer #2: Yes

5. Is the manuscript presented in an intelligible fashion and written in standard English?

Reviewer #1: Yes

Reviewer #2: Yes

6. Review Comments to the Author

Reviewer #1: The study has a good objective but the measures are not good (the CAT) and there are some questionable decisions (not examining all DT indices). I suppose researchers may learn something from this project, but it disappoints me that relevant research was not cited (e.g., Wallach and Wing) in the introduction and that the methodological issues mentioned earlier and above were not addressed.

Reviewer #2: As the reviewer's comments were considered fully and answered thoroughly, in my perspective the paper is acceptable now for publication. Well done!

7. PLOS authors have the option to publish the peer review history of their article (what does this mean?). If published, this will include your full peer review and any attached files.

Reviewer #1: No

Reviewer #2: No

---

## [Author Response · Author response to Decision Letter 1]

21 Apr 2022

We appreciate that the editor and both reviewers carefully checked our manuscript and provided us suggestive comments. According to the reviewers’ suggestions, we added more detailed descriptions in our manuscript. We would appreciate if you find below our responses to each comments.

Reviewer #1: 

The study has a good objective but the measures are not good (the CAT) and there are some questionable decisions (not examining all DT indices). I suppose researchers may learn something from this project, but it disappoints me that relevant research was not cited (e.g., Wallach and Wing) in the introduction and that the methodological issues mentioned earlier and above were not addressed.

Response. As you pointed out in the first peer review, unreliable indicators could lead to regression dilution and reduction in testing power (Spearman, 1904). Since it is clearly statistically inappropriate to use unreliable indicators for regression analysis, Haiku CAT could not be used in the subsequent analysis. Similarly, flexibility and originality of DT could not be included in the regression analysis due to reliability issues. Further, we found that the fluency and flexibility were highly correlated. Therefore, they could not be included in the regression analysis simultaneously. However, as your suggestion, it is needed to report about all sub-indicators of TTCT (Torrance, 1979; Kim, 2008). Therefore, we have added the results of Elaboration to the text. Because of the low reliability coefficients for flexibility and originality, the results of the analysis are presented in the Supporting information without mentioning them in the main text.

p. 13, l. 260: Since the use of the total score of the TTCT is controversial and reporting of all four sub-scales is encouraged [31,49], the current study first assessed the reliability of scoring for each aspect in the current sample and then examined the relationships among these aspects. A reliability score of greater than .80—or .70 in psychometric assessment—is required to avoid regression dilution and reduction in testing power [50,51]. Thus, the current results showed that the reliability scores (Cronbach’s alpha) of fluency and elaboration were adequate, whereas those of other evaluation aspects were not (fluency: .80; flexibility: .67; originality: .56; elaboration: .77). Furthermore, the correlation between fluency and elaboration was relatively high (r(86) = .889, p < .0001). Therefore, as the subsequent analyses could not include DT sub-scales simultaneously, the scores for fluency and elaboration were used individually because these were the most reliable and such aspects demonstrated similar characteristics to the others. The results of the analysis of flexibility and originality, which are not mentioned in the text due to their low reliability, are included in the supporting information (Table S1, Table S2, Table S3, Table S4).

p. 17, l. 322: We found similar results using elaboration as creative potential. A multiple regression analysis was conducted using creative production of art as the objective variable and elaboration, CPS score, and the interaction term between elaboration and CPS score as explanatory variables (Table 4). In Step 1, we found that elaboration had a positive main effect on creative production (β = .31, t(84) = 3.0, p = .004). In Step 2, CPS score was included as an explanatory variable and the effect of elaboration remained (β = .30, t(83) = 2.9, p = .004), but CPS exhibited no effect (β = .18, t(83) = 1.8, p = .080). In Step 3, the interaction term between the creative potential (elaboration) and CPS score was included as an explanatory variable. However, there remained no interaction effect between elaboration and CPS score (β = -.11, t(82) = -1.1, p = .294).

p. 18, l. 336: Further, a multiple regression analysis was conducted with CAQ score as the objective variable and elaboration, CPS score, and the interaction term between the elaboration and CPS score as explanatory variables (Table 5). In Step 1, we found that elaboration had a positive main effect on CAQ score (β = .23, t(85) = 2.2, p = .031). The results of the analysis in Step 2, in which CPS score was included as an explanatory variable, showed a positive effect of elaboration (β = .23, t(84) = 2.2, p = .031) and CPS score (β = .23, t(84) = 2.2, p = .032). In Step 3, the interaction term of elaboration and CPS score was included as an explanatory variable, which showed an interaction effect between elaboration and the CPS score (β = .39, t(83) = 4.0, p < .0001). The simple main effect test results showed that elaboration had a positive effect on CAQ score in the high CPS group (β = .70, t(83) = 4.6, p < .0001, Fig 5B) wever, elaboration exhibited no effect in the low CPS group (β = -.09, t(83) = -0.7, p = .482).

p. 24, l. 466: Although it is standard to report on all sub-indicators of TTCT [31,49], originality and flexibility were not included in the current analyses due to inadequate reliability. Since these are two core concepts in DT, further research is necessary to examine whether these aspects produce similar results. The low reliabity of these sub-indicators was caused by issues with the Japanese TTCT scoring process.

We appreciate that your comments enabled us to learn a lot about creativity measurements. In fact, we have fully realized that the measurements in our study have a lot of limitations. Therefore, we demonstrated the limitations in the last amendments according to your comments. However, as you suggested, our description did not sufficiently address how researchers should measure creativity in the future study. Therefore, we have added a more detailed description about it as our maximum possible response in this manuscript. In addition, based on the limitations of this research, we are now starting a new project to review the issues in Japanese creativity research and introduce appropriate scoring flows for creative potential (e.g., Reiter-Palmon et al., 2019) and to develop a scoring system with applying natural language processing. We will build the basis of creativity research with appropriate procedures.

p. 24, l. 466: Although it is standard to report on all sub-indicators of TTCT [31,49], originality and flexibility were not included in the current analyses due to inadequate reliability. Since these are two core concepts in DT, further research is necessary to examine whether these aspects produce similar results. The low reliability of these sub-indicators was caused by issues with the Japanese TTCT scoring process. The study scales were translated and redeveloped from the TTCT for Japan. However, we outsourced scoring to an external organization. Thus, the scoring process was closed. Tests like the TTCT, translated worldwide, are convenient and easy to use. Still, researchers should pay attention to the validity and reliability of the scoring process in each test. Future studies in Japan should utilize a new scoring system—as described below—to ensure reliable and valid scoring.

p. 25, l. 484: Recently, researchers have indicated a procedural issue regarding creativity measurement in which scorers evaluate ideas or products. In scoring DT tasks or the CAT, the scorer should be trained in DT scoring [63] or have sufficient knowledge and experience in a particular creative domain [3]. Studies have shown that despite recruiting such scorers, their scoring generally became burdensome, especially with a large sample [64,65]. To solve these issues, new scoring systems using Latent Semantic Analysis (LSA) [66] have been proposed in recent creativity research [67,68]. LSA scoring has the potential to enable researchers to automatically score DT or literal products with high reliability and validity [64]. As such scoring systems have not yet been proposed in Japanese, future studies are necessary to develop these systems and conduct creativity research based on reliable and valid Japanese measurements.

Although we cited the Wallach & Wing(1969) in the introduction in the previous revision, the description may not have been able to adequately account for their contributions. In this paper, we have added a specific description of the Wallach & Wing (1969) study in the introduction.

p. 3, l. 49: However, this association between creative potential and performance is weakly positive in most studies conducted on children to adults [13–17]. The predictive role of creative potential has also been examined but remains controversial. Wallach and Wing [15] showed a positive association between creative potential and creative performance. They conducted an experimental survey of 503 freshman university students and found that creative potential predicted non-academic outcomes that IQ and standardized tests alone did not. Subsequent longer-term longitudinal studies have examined the predictive power of DT in more detail. Kogan and Panvoke [13] showed that children’s creative potentials correlated with their accomplishments in the 5th but not in the 10th grade. The 18-year follow-up study also showed that the creative potential of high school students was only associated with specific areas of creative achievements [18]. Torrance [10] designed the Torrance Test of Creative Thinking (TTCT) to indicate creative potential that promotes creative behaviors. In addition, previous empirical research with a 40- or 50-year follow-up found that creative ideation scores predict future creative achievement [19-21]. Further, it is suggested in a meta-analysis [22] and follow-up study [19] that the TTCT has a better predictive association of creative achievements than IQ. However, it has been argued that the link between creative potential and behavior is weak (r score is less than .30) and sometimes insignificant [23]. An examination of what causes this weak correlation follows.

---

## [Decision Letter · Decision Letter 2]

20 Jun 2022

PONE-D-21-18303R2Relationships among creativity indices: Creative potential, production, achievement, and personalityPLOS ONE

Dear Dr. Takagishi,

Thank you for submitting your manuscript to PLOS ONE. After careful consideration, we feel that it has merit but does not fully meet PLOS ONE’s publication criteria as it currently stands. Therefore, we invite you to submit a revised version of the manuscript that addresses the points raised during the review process.

The new reviews indicate a potentially unsurpassable set of issues with the study reported in the paper: 1. sample size is very small 2. the use of CPS as a personality measureIn your second revision of the manuscript please consider all the comments provided by the latest two reviewers and also indicate the limitations above in the abstract and discussion section. Please submit your revised manuscript by Aug 04 2022 11:59PM. If you will need more time than this to complete your revisions, please reply to this message or contact the journal office at plosone@plos.org. Please include the following items when submitting your revised manuscript:A rebuttal letter that responds to each point raised by the academic editor and reviewer(s). You should upload this letter as a separate file labeled 'Response to Reviewers'.A marked-up copy of your manuscript that highlights changes made to the original version. You should upload this as a separate file labeled 'Revised Manuscript with Track Changes'.An unmarked version of your revised paper without tracked changes. You should upload this as a separate file labeled 'Manuscript'.If applicable, we recommend that you deposit your laboratory protocols in protocols.io to enhance the reproducibility of your results. Protocols.io assigns your protocol its own identifier (DOI) so that it can be cited independently in the future. For instructions see: https://journals.plos.org/plosone/s/submission-guidelines#loc-laboratory-protocols. Additionally, PLOS ONE offers an option for publishing peer-reviewed Lab Protocol articles, which describe protocols hosted on protocols.io. Read more information on sharing protocols at https://plos.org/protocols?utm_medium=editorial-email&utm_source=authorletters&utm_campaign=protocols.

We look forward to receiving your revised manuscript.

Kind regards,

Denis Alves Coelho, PhD

Academic Editor

PLOS ONE

Journal Requirements:

Reviewers' comments:

Reviewer's Responses to Questions

**Comments to the Author**

1. If the authors have adequately addressed your comments raised in a previous round of review and you feel that this manuscript is now acceptable for publication, you may indicate that here to bypass the “Comments to the Author” section, enter your conflict of interest statement in the “Confidential to Editor” section, and submit your "Accept" recommendation.

Reviewer #3: (No Response)

Reviewer #4: All comments have been addressed

2. Is the manuscript technically sound, and do the data support the conclusions?

Reviewer #3: Partly

Reviewer #4: Yes

3. Has the statistical analysis been performed appropriately and rigorously? 

Reviewer #3: Yes

Reviewer #4: Yes

4. Have the authors made all data underlying the findings in their manuscript fully available?

Reviewer #3: (No Response)

Reviewer #4: Yes

5. Is the manuscript presented in an intelligible fashion and written in standard English?

Reviewer #3: Yes

Reviewer #4: Yes

6. Review Comments to the Author

Reviewer #3: The paper addresses a major issue in the study of creativity, that of measurement and the inter-relationships between different measures of creativity. The authors note that different measures likely measure different aspects of creativity, and that the relationships between them is not always understood. While the authors attempt to clarify some issues, I do have a number of concerns. I also understanding that I am coming in later in the review process, with the authors already revising their paper in response to previous reviewer comments. I have taken the approach of reading the paper as is and responding to that, and my apologies if I am asking you to go back on some previous comments!

1. The justification for personality as a moderating variable is not clear to me. One issue I have is the use of the word “intervene” which to me suggests mediation and not moderation. Further, the description of the evidence does not necessarily suggest that personality is a moderator. This part of your introduction needs to be strengthened. I would also suggest that you focus specifically on the measure of personality or personality aspects you are including in your study. Personality can include multiple variables, and there is no reason to believe that findings from one personality characteristic would apply to another.

2. Justification for the selection of CPS as a personality variable is also limited. Just because the research has not been conducted, does not mean that it should. You should justify why that is a gap in our understanding of the link between personality and creativity.

3. Further, the CPS as a measure of personality includes multiple personality traits, and as such is more difficult to interpret. This makes supporting its use and interpretation of the results more difficult. Using specific and more targeted personality variables (like the big 5, or specific traits such as tolerance for ambiguity or creative self-efficacy) would provide a better understanding of the relationship between personality and creativity.

4. In part you reason that you choose two different production tasks – one in visual art and one in writing. I will note a few concerns. First, due to low reliability, you had to omit one of the tasks. Second, both tasks are in the art domain. I would expect a much different relationship may emerge if you used very different domains such as everyday problem solving and art. One solution to both of these issues is to avoid the issue of multiple domains in the introduction, and note the concern of domain specificity in the discussion.

5. The last section in the introduction about neuroscience seems out of place and not really relevant to the study. Clarity regarding measurement is critical for the field as a whole.

6. Your sample size is too small. Research indicates that Correlational (and regression by extension) research does not provide stable results until 150-250 participants (see Schönbrodt & Perugini, 2013).

7. You discuss domain issues, but even if you could test on both production tasks, the two are both art based(writing a haiku and cut out). In that respect, they are focused on two sub-domains within the art domain and the argument throughout for the importance of domains seems misaligned.

8. There are a number of issues with your discussion of scoring of the different tasks.

a. Please provide additional clarity on the ratings of the divergent thinking tasks. You mentioned that they were scored by expert judges, and later by a professional organization. Is this the same or different? Do you have information on inter-rater reliability for this?

b. Inter-rater reliability should be evaluated using Interclass Correlations (ICC, Shrout and Fleiss, 1979). Alphas are akin to ICC (3) and tend to be a more inflated measure of inter-rater reliability. The number of raters (5) also influences this. Therefore, an inter-rater agreement of .72 for the CAT is on the low end.

c. I found the description of the CAT ratings confusing. Were ratings given on each dimension? If not, what were the 10 dimensions used for? How were the 10 dimensions used to create a final score? Was reliability calculated across or within each dimension? Please provide more detailed information about the rating and scoring.

9. You state that you used the total CAQ score. There is a debate about whether the total should be used at all (see Silvia et al., 2012). Using the total score would require a justification which you currently do not provide. You could alternatively use the factor solution suggested by Carson and he colleagues to create 2 larger scales.

10. Your discussion about the total creativity score and summing was confusing. Since you did not do that, I would remove the mention of the sum, and just focus on the fact that you did not sum the scores because it is not appropriate to do so.

11. Some aspects of your discussion section are better suited for the introduction. For example, the discussion about conceptualizations of creativity fits better in the introduction. Specifically, a more detailed discussion the 4Ps and how they relate to the current study should be included in the introduction. The discussion should reiterate this but in a brief sentence.

12. Language considerations – you seem to use CAT and production task interchangeably, the same applies to DT/TCTT/creative potential. The first is confusing because CAT is the rating process not the task. The second is confusing because creative potential is the theoretical concept and DT or TCTT are the tasks used to measure it. Please be consistent in their use in the appropriate place.

There is a typo in author names for citation #37.

References

Schönbrodt & Perugini (2013). At what sample size do correlations stabilize?, Journal of Research in Personality, 47, 609-612. https://doi.org/10.1016/j.jrp.2013.05.00

Reviewer #4: Generally speaking, this article is quite good. I have however a few concerns.

1. At a theoretical level, I think the paper suffer several weaknesses (see detail below).

2. I think that it is problematic to call “creative potential” something that, in fact, boils down to divergent thinking. Creative potential is a broader concept than divergent thinking, see e.g. Lubart et al. 2013 https://files.eric.ed.gov/fulltext/EJ1301375.pdf

3. There have also been recent theoretical and empirical proposals concerning the articulation between personality, process and creative achievement, also in connection with the notion of creative potential, see e.g. Fürst & Grin 2018 https://www.sciencedirect.com/science/article/abs/pii/S1871187117303334

4. Major works in creative personality were also done by Hans Eysenck, see e.g. the book Genius: The Natural History of Creativity or Eysenck 1993 https://www.tandfonline.com/doi/abs/10.1207/s15327965pli0403_1

5. When speaking of creative personality, it also seems essential to talk about the personality factor Openness, which is the main factor associated to personality. This seems especially crucial because the CPS is arguably a measure of Openness. (Most items of this scale are very similar to the key items of Openness, e.g. “Love to daydream”, “Like to solve complex problems”, “Have a vivid imagination”, etc.).

6. At the empirical level, the analysis reported in Table 2-3 and Table 4-5 seem redundant. It is the same results, once with Fluency and once with Elaboration. I would have simply used a total divergent thinking score combining fluency, flexibility, originality and elaboration. I understand that the authors were reluctant to do this. But still, I wonder what is the correlation between fluency and elaboration? And is this really necessary to perform two analysis that basically return the same results?

7. In line with previous comment, I think that it would be a good thing to provide a correlation matrix of all variables in the study. Some of them are reported in the text, but it is not exhaustive and it is not convenient.

8. P. 22, the authors mention the Runco Ideationnal Behavior Scale as a measure of creative achievement. This scale is not a measure of creative achievement, it is a measure of ideational “habits”, no item of this scale is focused on achievement whatsoever.

9. At the very end of the paper, the authors say that future research should clarify the influence of personality on creativity. The literature on this topic is so vast and so much work have been done in this direction that such a comment is simply not acceptable as a way to close the paper!

7. PLOS authors have the option to publish the peer review history of their article (what does this mean?). If published, this will include your full peer review and any attached files.

Reviewer #3: No

Reviewer #4: No

---

## [Author Response · Author response to Decision Letter 2]

5 Aug 2022

Response to Reviewers’ comments

We appreciate that the editor and both reviewers carefully checked our manuscript and provided us with suggestive comments. According to the reviewers’ suggestions, we added more detailed descriptions to our manuscript. We would appreciate it if you find below our responses to each comment.

Reviewer #3: 

Comment 1. The justification for personality as a moderating variable is not clear to me. One issue I have is the use of the word “intervene” which to me suggests mediation and not moderation. Further, the description of the evidence does not necessarily suggest that personality is a moderator. This part of your introduction needs to be strengthened. I would also suggest that you focus specifically on the measure of personality or personality aspects you are including in your study. Personality can include multiple variables, and there is no reason to believe that findings from one personality characteristic would apply to another.

Response. The last manuscript was vague in the justification for the role of the creative personality. As your suggestion, we avoided using the “intervene” and similar expressions. Further, we added the description that the current study focuses on CPS as individuals’ beliefs on creative personality and how the beliefs on creative personality affect the association between DT and creative production/achievements. Focusing on the role of beliefs in creative personality, we assumed that the beliefs could have mediating or moderating role between DT and creative production/achievements according to CBAA theory and its empirical findings. As creative personality can be considered as creative potential (Lubart et al., 2013), it can be assumed that the creative personality beliefs cannot be the antecedents of DT, which is assumed as one of creative potential, too. Thus, we assumed the moderating role of creative personality beliefs on the relationship between DT and creative production/achievements.

As you mention, personality can have multiple variables. Although the current study applied CPS to capture the beliefs of various creative personalities, the findings cannot be applied to the other personalities. Thus, the discussion about the role of personality was amended in the current manuscript.

P. 6, l. 109: Here, a question is begged: what can explain these findings on the inconsistent and weak association between DT and creative production and achievements? A recent theory by Karwowski and Beghetto [38] provides an outlook on this matter, proposing that individuals actualize their creative potential based on their personal intentions. They named the theory Creative Behavior as an Agentic Action (CBAA), positing that creative self-beliefs (which encompasses creative self-efficacy and perceived value of creativity) is a personal factor that mediates (through creative self-efficacy) and moderates (through perceived value of creativity) the relationship between DT and creative behavior [38, 39]. Hence, CBAA proposes that individual traits and attitudes toward creativity can influence the relationship between DT and creative behavior. Importantly, these propositions of CBAA raise the question as to whether other characteristics at the individual level can influence this relationship.

Kaufman [40] suggested that self-assessed measures of creative personality (e.g., openness to experience and beliefs about own creative personality) can be used to perceive how individuals view their creativity in addition to measures of creative self-beliefs (e.g., creative self-efficacy and perceived value of creativity). That is, beliefs about own creative personality can have a mediating or moderating role on the association of DT with creative production and achievement. Since these trait-related measures of creativity are assumed to be conative dimensions of creative potential [14], they cannot be preceded by DT, namely, they cannot have a meditating role on the relation between DT and creative behavior. Here, an interrogation arises: can beliefs about own creative personality strengthen individuals’ actualization of DT into creative behaviors? The confirmation of such an assumption may provide support to the triangle relationship of the Person, Process, and Product dimensions.

P. 22, l. 412: According to the assumption by Lubart et al. [14], creative potential includes cognitive resources (e.g., DT) and conative resources (e.g., openness to experiences), and used DT and creative personality (openness) as psychological components. Then, these cited authors constructed a latent factor of creative potential by combining DT and openness, showing their impacts on creative production and achievements. This type of functional model of creative personality comprising DT and production and achievements is important when focusing on personalities, as personalities are considered to have close relationships with genetic and other innate factors (e.g., openness to experiences and novelty seeking) [59,60]. Meanwhile, when measuring personality traits using self-assessed scales, the personality scores show individuals’ own beliefs about each item. If we consider that the CPS measures beliefs about own creative personality, another function of conative resources of creative potential comes to light: Beliefs about own conative resources influence how DT is realized in actual creative behavior. Future research should examine the dual function of creative personality creative personality and beliefs about own creative personality in creativity actualization, as this may yield relevant data for better understanding the specific function of the Person dimension in the 4 Ps theory.

Comment 2. Justification for the selection of CPS as a personality variable is also limited. Just because the research has not been conducted, does not mean that it should. You should justify why that is a gap in our understanding of the link between personality and creativity.

Response. The last manuscript did not fully explain why CPS should be applied to the current investigation. The reason why CPS was used in the current study is that the current study aimed to focus on the beliefs of a wide range of creative personalities. Since CPS was developed from the items related to creativity but not from the previous personality research, thus it covers a wider range of creative personality traits. This justification was described in the introduction section.

P. 7, l. 139: We used the self-assessed Creative Personality Scale (CPS) [41, 42] to examine beliefs about own creative personality because it comprises items that relate to various creative traits of thinking, problem-solving, and imagination. Although the CPS does not specifically assess personality, it does yield self-assessed data on a wide range of creative traits, allowing us to understand the moderating role of beliefs about own creative personality on the dimensions of creativity.

Comment 3. Further, the CPS as a measure of personality includes multiple personality traits, and as such is more difficult to interpret. This makes supporting its use and interpretation of the results more difficult. Using specific and more targeted personality variables (like the big 5, or specific traits such as tolerance for ambiguity or creative self-efficacy) would provide a better understanding of the relationship between personality and creativity.

Response. The last manuscript did not explain the reason for using CPS other than openness or other creative personality. The current manuscript demonstrated that the current study applied CPS as the belief of creative personality because it was developed from personality items that relate to creativity. Of course, even if we applied CPS as beliefs of creative personality, the function of other personalities should be examined. Especially, personalities such as openness to experience and novelty seeking are important ones because they are closely related to genetic or innate factors. Thus, we described that the functions of such personality traits and beliefs of specific creative personalities should be further investigated in the future study in the Discussion section.

P. 22, l. 412: According to the assumption by Lubart et al. [14], creative potential includes cognitive resources (e.g., DT) and conative resources (e.g., openness to experiences), and used DT and creative personality (openness) as psychological components. Then, these cited authors constructed a latent factor of creative potential by combining DT and openness, showing their impacts on creative production and achievements. This type of functional model of creative personality comprising DT and production and achievements is important when focusing on personalities, as personalities are considered to have close relationships with genetic and other innate factors (e.g., openness to experiences and novelty seeking) [59,60]. Meanwhile, when measuring personality traits using self-assessed scales, the personality scores show individuals’ own beliefs about each item. If we consider that the CPS measures beliefs about own creative personality, another function of conative resources of creative potential comes to light: Beliefs about own conative resources influence how DT is realized in actual creative behavior. Future research should examine the dual function of creative personality creative personality and beliefs about own creative personality in creativity actualization, as this may yield relevant data for better understanding the specific function of the Person dimension in the 4 Ps theory.

Comment 4. In part you reason that you choose two different production tasks – one in visual art and one in writing. I will note a few concerns. First, due to low reliability, you had to omit one of the tasks. Second, both tasks are in the art domain. I would expect a much different relationship may emerge if you used very different domains such as everyday problem solving and art. One solution to both of these issues is to avoid the issue of multiple domains in the introduction, and note the concern of domain specificity in the discussion.

Response. The last manuscript did not provide a valid solution to clarify whether or not there is a difference in the associations between the beliefs on creative personality, DT, and production/achievements across specific creative domains. Thus, as you suggested, the current manuscript avoided to emphasize the domain specificity problems in the introduction section. Otherwise, we discussed the necessity to examine the domain-specificity problem in the discussion section. 

P. 23, l. 433: Furthermore, there are other creative domains (e.g., science, social activities, etc.) which have yet to be examined by scholars. Thus, future studies should explore the differences in the association between DT and creative production across different creative domains. Interestingly, the moderating role of beliefs about own creative personality was not consistent between creative behaviors (i.e., production and achievement). Instead, beliefs about own creative personality did not explain the increase in variance in creative production, nor was it significantly associated with creative production in fine arts. This low explanation from beliefs about own creative personality may be due to the settings of the creative production tasks proposed in the current study, which only dealt with an art domain. Accordingly, although the participants in this study had higher creative personality, they may not have been willing to utilize their DT in a cutouts task. In other words, beliefs about own creative personality regarding specific creative domains are likely to influence DT differently across tasks in different creative domains. Kaufman and Baer [42] showed a significant relationship between CPS and self-assessed creativity in specific creative domains, albeit the relationships were weak (e.g., r = .23 in the art domain). Thus, future research should focus on beliefs about own creative personality in specific creative domains when making domain-specific examinations of the impact of such beliefs on the relationship between DT and creative production.

Comment 5. The last section in the introduction about neuroscience seems out of place and not really relevant to the study. Clarity regarding measurement is critical for the field as a whole.

Response. As your suggestion, the current study itself was not related to neuroscience. Thus, the current manuscript excluded the neuroscience parts.

Comment 6. Your sample size is too small. Research indicates that Correlational (and regression by extension) research does not provide stable results until 150-250 participants (see Schönbrodt & Perugini, 2013).

Schönbrodt & Perugini (2013). At what sample size do correlations stabilize?, Journal of Research in Personality, 47, 609-612. https://doi.org/10.1016/j.jrp.2013.05.00

Response. As you noted, the number of participants in this study (n = 88) may not be large. However, when we performed a post-hoc power analysis using G*Power 3.1, all of our main results exceeded a power of 0.8. These results indicate that 88 participants are not a small number. We have added the results of the sensitivity power analysis to the main text.

This study involved creative production tasks in two areas that were time and labor intensive for the participants. In addition, given the effort required to evaluate the experts, it was not possible to increase the number of participants in this study. However, as you noted, larger sample size is necessary for the stability of the correlation scores. Therefore, we have added this issue as a limitation of this study.

P. 13, l. 273: This study did not use a prior sample design. Therefore, we conducted a sensitivity power analysis using G*Power 3.1 [53]. For a correlation analysis between two variables with power set at 0.8, we could theoretically detect an effect size (ρ) greater than 0.289 with 88 participants, and greater than 0.291 with 87 participants. 

Regarding moderation effect analyses, with power set at 0.8, we could theoretically detect an effect size (f2) greater than 0.091 with 88 participants, and an effect size (f2) greater than 0.092 with 87 participants.

P. 28, l. 547: Finally, although we tried to recruit the largest sample possible, we were not able to recruit more than 100 participants due to budget constraints and the potential burdens on participants and creative production task raters. Accordingly, although the current sample size (N = 88) has sufficient statistical power, it did not show sufficient stability for the estimates according to research by Schönbrodt and Perugini [73]; these authors posit that correlational studies should collect data from more than 150 participants for ensuring estimate stability. Thus, future researchers should try and replicate the current research with larger samples to confirm our findings.

Comment 7. You discuss domain issues, but even if you could test on both production tasks, the two are both art based(writing a haiku and cut out). In that respect, they are focused on two sub-domains within the art domain and the argument throughout for the importance of domains seems misaligned.

Response. As you suggest, the last manuscript did not provide a valid solution to the domain specificity problem because the current study conducted creative production tasks only in two art domains. Thus, as you suggested, the current manuscript avoided focusing on the domain specificity problems in the introduction section. Otherwise, we discussed the necessity to examine whether or not there is a difference in the associations of beliefs on creative personality, DT, and production/achievements across different creative domains such as engineering, science, or leadership other than arts.

P. 23, l. 433: Furthermore, there are other creative domains (e.g., science, social activities, etc.) which have yet to be examined by scholars. Thus, future studies should explore the differences in the association between DT and creative production across different creative domains. Interestingly, the moderating role of beliefs about own creative personality was not consistent between creative behaviors (i.e., production and achievement). Instead, beliefs about own creative personality did not explain the increase in variance in creative production, nor was it significantly associated with creative production in fine arts. This low explanation from beliefs about own creative personality may be due to the settings of the creative production tasks proposed in the current study, which only dealt with an art domain. Accordingly, although the participants in this study had higher creative personality, they may not have been willing to utilize their DT in a cutouts task. In other words, beliefs about own creative personality regarding specific creative domains are likely to influence DT differently across tasks in different creative domains. Kaufman and Baer [42] showed a significant relationship between CPS and self-assessed creativity in specific creative domains, albeit the relationships were weak (e.g., r = .23 in the art domain). Thus, future research should focus on beliefs about own creative personality in specific creative domains when making domain-specific examinations of the impact of such beliefs on the relationship between DT and creative production.

Comment 8. There are a number of issues with your discussion of scoring of the different tasks.

a. Please provide additional clarity on the ratings of the divergent thinking tasks. You mentioned that they were scored by expert judges, and later by a professional organization. Is this the same or different? Do you have information on inter-rater reliability for this?

Response. The last manuscript was unclear about DT scoring process. The DT rating was performed by one judge from a professional organization. The organization does not open the specific procedure of the scoring. However, according to the communication with the organization, in the scoring, one expert judge rated the participants’ responses in accord with the scoring chart created from sample data of the Japanese population. Thus, the organization did not provide inter-rater reliability information. This information and further specification of the procedure were added in the method section.

P. 10, l. 199: The test is scored based on four aspects: fluency, flexibility, originality, and elaboration. Fluency is a measure of the ability to generate many ideas, being evaluated by the number of responses excluding those that are inappropriate or difficult to interpret. Flexibility is the ability to generate ideas from a wide range of perspectives, being assessed by the number of categories in the ideas generated and according to a criterion table (i.e., indicating a particular answer’s classification and how many points it receives) or equivalent judgment [43]. Originality is the ability to generate ideas different from those of others, which is again evaluated based on a criterion table (i.e., indicating the frequency of occurrence of categories for each response) [43]. Those with a frequency of occurrence of less than 1% scored 2 points, of 1–5% scored 1 point, and of more than 1% scored 0 points [43]. Finally, elaboration is the ability to think concretely about ideas, being evaluated based on the total number of responses weighted by a criterion table (i.e., describing how well the response depicts the means and structure of the purpose or functions) or equivalent judgments [43]. Torrance [47] initially established these four dimensions for evaluating ideas based on the elements of DT as proposed by Guilford [13]. 

One trained judge from an external professional organization performed all of the ratings; the expert judge evaluated the participants’ responses while following the scoring charts developed according to data from a sample of the Japanese population (Tokyo Shinri Corporations) [43].

b. Inter-rater reliability should be evaluated using Interclass Correlations (ICC, Shrout and Fleiss, 1979). Alphas are akin to ICC (3) and tend to be a more inflated measure of inter-rater reliability. The number of raters (5) also influences this. Therefore, an inter-rater agreement of .72 for the CAT is on the low end.

Response. As you suggest, the ICC is a more appropriate measure to assess the raters’ agreement. Thus, we re-calculated intraclass-correlations, following the Cseh et al. (2019), Koo & Li (2016), and Shrout & Fleiss(1979). The current study assumed that each participant’s product was rated by a different 5 raters from a larger sample and the raters’ scores will be calculated into average scores. Therefore, the ICC (2, k) (Shrout & Fleiss, 1979; Koo & Li, 2016) was calculated using “irr” package for R (Gamer et al., 2019) based on a mean-rating (k = 5), absolute-agreement, 2-way mixed-effects model, which was an acceptable level of agreement. This procedure was applied to calculate inter-rater agreements in both of Haiku and cutout pieces evaluation. As the same as the last analyses, Haiku evaluation agreements were quite poor. In regard to the cutout pieces, as you noted, ICC (3, k) (which is the same as alpha) was high (.80) and ICC(2,k), which assumes absolute agreement, was smaller (0.75). However, we still observed a moderately high rate of rater agreement.

P. 12, l. 243: Following Amabile’s [20] procedure, factor analyses were performed for rating three Haikus of each participant by five raters. The eigen values (4.15, 1.08, 0.19, -0.02) suggested that the dimensions consisted of two factors and the reliabilities as a consistency were good (αs> .80). Further, the CAT scoring procedure requires inter-rater agreement [48]; although the current study calculated ICC(2, k) using the “irr” package for R [49] based on a mean-rating (k = 5), absolute-agreement, and 2-way mixed-effects model, the inter-rater agreements were quite poor (ICCs(2,k) < .20). Thus, the results of the Haiku ratings were not included for further analysis. (… ) Factor analyses were performed for rating two products by each participant by five raters. The eigen values (5.72, 0.46, 0.21, 0.08) suggested that the dimensions consisted of one factor and the reliability as a consistency was good (α= .95). Further, the CAT scoring procedure requires the inter-rater agreement [48]; the current study calculated ICC(2, k) into 0.75 [CI: 0.62-0.84] based on a mean-rating (k = 5), absolute-agreement, and 2-way mixed-effects model, which was an acceptable level of agreement.

c. I found the description of the CAT ratings confusing. Were ratings given on each dimension? If not, what were the 10 dimensions used for? How were the 10 dimensions used to create a final score? Was reliability calculated across or within each dimension? Please provide more detailed information about the rating and scoring.

Response. The procedure of CAT rating was unclear in the last manuscript. The expert judges rated each product using 10 dimensions respectively. According to Amabile (1983)’s CAT procedure, factor analyses were performed on the scores of two participants’ works rated by five raters on 10 dimensions. The results indicated the one factor solution; therefore, the subsequent analyses used the average score of 10 dimensions as the artistic production score. These procedures were described more specifically in the results section.

P. 13, l. 256: Factor analyses were performed for rating two products by each participant by five raters. The eigen values (5.72, 0.46, 0.21, 0.08) suggested that the dimensions consisted of one factor and the reliability as a consistency was good (α= .95). Further, the CAT scoring procedure requires the inter-rater agreement [48]; the current study calculated ICC(2, k) into 0.75 [CI: 0.62-0.84] based on a mean-rating (k = 5), absolute-agreement, and 2-way mixed-effects model, which was an acceptable level of agreement.

Comment 9. You state that you used the total CAQ score. There is a debate about whether the total should be used at all (see Silvia et al., 2012). Using the total score would require a justification which you currently do not provide. You could alternatively use the factor solution suggested by Carson and he colleagues to create 2 larger scales.

Response. As your suggestion, we could have analyzed the factor structure of CAQ score; however, the sample size of the current study is only 88, which is below the sample size required in the factor analyses (e.g., Kline, 1993; COSMIN checklist). Therefore, the current study could not implement the factor analysis. 

Comment 10. Your discussion about the total creativity score and summing was confusing. Since you did not do that, I would remove the mention of the sum, and just focus on the fact that you did not sum the scores because it is not appropriate to do so.

Response. This discussion was added according to the previous discussion with the reviewers. In fact, the S-A creativity test organization provided the total score. Further, in the current review process, the other reviewer also suggested using the total score of DT. However, the previous studies discouraged to use of the total score of DT, we decided to describe our policy to analyze DT scores in the result section.

Comment 11. Some aspects of your discussion section are better suited for the introduction. For example, the discussion about conceptualizations of creativity fits better in the introduction. Specifically, a more detailed discussion the 4Ps and how they relate to the current study should be included in the introduction. The discussion should reiterate this but in a brief sentence. 

Response. The last manuscript did not fully demonstrate the typology of creativity. According to your suggestion and the other reviewer’s comments, the current manuscript added a more detailed discussion about 4Ps in the introduction and make the discussion section brief. 

Comment 12. Language considerations – you seem to use CAT and production task interchangeably, the same applies to DT/TCTT/creative potential. The first is confusing because CAT is the rating process not the task. The second is confusing because creative potential is the theoretical concept and DT or TCTT are the tasks used to measure it. Please be consistent in their use in the appropriate place.

Response. As you suggest, the last manuscript had the confusion in wording of concepts related to creativity (e.,g DT, creative production/achievement) and creative measures (e.g., S-A creativity test, creative production score, CAQ score). Thus, the current manuscript amended these confusing words thoroughly.

Comment 13. There is a typo in author names for citation #37.

Response. We amended the typo.

 

Reviewer #4: 

Comment 1. At a theoretical level, I think the paper suffer several weaknesses (see detail below).I think that it is problematic to call “creative potential” something that, in fact, boils down to divergent thinking. Creative potential is a broader concept than divergent thinking, see e.g. Lubart et al. 2013 https://files.eric.ed.gov/fulltext/EJ1301375.pdf

Response. We appreciate that you provide the references. As you suggest, Lubart et al. (2013) suggested that creative potential includes various components such as divergent thinking or creative personality. Therefore, we amended the way to use the term “creative potential” in the current manuscript thoroughly. Further, this theoretical classification was demonstrated in the introduction discussion about 4Ps.

P. 3, l. 54: Creative potential, in turn, is expected to relate to and predict creative production and achievements, which relate to the Product dimension [16]. Following these theoretical assumptions, various researchers have examined the impact of creative personality (Person dimension) and creative thinking (Process dimension) on creative production and achievements (Product dimension). However, the relationships among these creativity indices remain unclear.

(…)

Since these trait-related measures of creativity are assumed to be conative dimensions of creative potential [14], they cannot be preceded by DT, namely, they cannot have a meditating role on the relation between DT and creative behavior.

P. 22, l. 412: According to the assumption by Lubart et al. [14], creative potential includes cognitive resources (e.g., DT) and conative resources (e.g., openness to experiences), and used DT and creative personality (openness) as psychological components. Then, these cited authors constructed a latent factor of creative potential by combining DT and openness, showing their impacts on creative production and achievements.

Comment 2. There have also been recent theoretical and empirical proposals concerning the articulation between personality, process and creative achievement, also in connection with the notion of creative potential, see 

e.g. Fürst & Grin 2018 https://www.sciencedirect.com/science/article/abs/pii/S1871187117303334

Response. We appreciate your providing further reference. This finding also supported us to understand the relationships between creative personality, DT, and production/achievements. We added a further discussion of the relationships by referring to this article in the discussion section.

P. 22, l. 411: For instance, Fürst and Grin used the score of openness/intellect as an index of creative personality [58]. According to the assumption by Lubart et al. [14], creative potential includes cognitive resources (e.g., DT) and conative resources (e.g., openness to experiences), and used DT and creative personality (openness) as psychological components. Then, these cited authors constructed a latent factor of creative potential by combining DT and openness, showing their impacts on creative production and achievements. This type of functional model of creative personality comprising DT and production and achievements is important when focusing on personalities, as personalities are considered to have close relationships with genetic and other innate factors (e.g., openness to experiences and novelty seeking) [59,60].

Comment 3. Major works in creative personality were also done by Hans Eysenck, see e.g. the book Genius: The Natural History of Creativity or 

Eysenck 1993 https://www.tandfonline.com/doi/abs/10.1207/s15327965pli0403_1

Response. We appreciate your providing further reference. This book was informative to understand the characteristics of personality in creative genius. As a result of amending the manuscript, the current study described our focus on creative personality as the beliefs on creative personality.

Comment 4. When speaking of creative personality, it also seems essential to talk about the personality factor Openness, which is the main factor associated to personality. This seems especially crucial because the CPS is arguably a measure of Openness. (Most items of this scale are very similar to the key items of Openness, e.g. “Love to daydream”, “Like to solve complex problems”, “Have a vivid imagination”, etc.).

Response. As you pointed out, the last manuscript did not describe the representative of creative personality such as openness to experience, tolerance of ambiguity, and so on. Thus, the current manuscript discriminated the difference between CPS and the other creative personality. This study focused on the CPS because it shows individuals’ beliefs on creative personality. Although CPS is, as you pointed out, quite a similar concept to openness, it covers the personality items that relate to creativity. Thus, it is useful to examine the role of beliefs of creative personality on the relationships between DT and creative production/achievements with applying CPS. However, this is the first step of examining the beliefs on creative personality role on the association between DT and creative production/achievements. Thus, the current manuscript demonstrated the necessity to further investigating the function of specific creative personality (i.e., openness to experience, tolerance of ambiguity and so on) in the discussion section. 

P. 22, l. 412: According to the assumption by Lubart et al. [14], creative potential includes cognitive resources (e.g., DT) and conative resources (e.g., openness to experiences), and used DT and creative personality (openness) as psychological components. Then, these cited authors constructed a latent factor of creative potential by combining DT and openness, showing their impacts on creative production and achievements. This type of functional model of creative personality comprising DT and production and achievements is important when focusing on personalities, as personalities are considered to have close relationships with genetic and other innate factors (e.g., openness to experiences and novelty seeking) [59,60]. Meanwhile, when measuring personality traits using self-assessed scales, the personality scores show individuals’ own beliefs about each item. If we consider that the CPS measures beliefs about own creative personality, another function of conative resources of creative potential comes to light: Beliefs about own conative resources influence how DT is realized in actual creative behavior. Future research should examine the dual function of creative personality creative personality and beliefs about own creative personality in creativity actualization, as this may yield relevant data for better understanding the specific function of the Person dimension in the 4 Ps theory.

Comment 5. At the empirical level, the analysis reported in Table 2-3 and Table 4-5 seem redundant. It is the same results, once with Fluency and once with Elaboration. I would have simply used a total divergent thinking score combining fluency, flexibility, originality and elaboration. I understand that the authors were reluctant to do this. But still, I wonder what is the correlation between fluency and elaboration? And is this really necessary to perform two analysis that basically return the same results

Response. In regard to the sum score, the previous study discouraged producing the total score of DT as described in the manuscript. Further, in the previous review process, the other reviewer also discouraged using the total score of DT. Thus, we decided to conduct the current analysis for each DT subscale and describe our policy to analyze DT scores in the result section. Thus, the Tables remained in the same manner.

Comment 6. In line with previous comment, I think that it would be a good thing to provide a correlation matrix of all variables in the study. Some of them are reported in the text, but it is not exhaustive and it is not convenient.

Response. As your suggestion, we added a correlation matrix in Table 2 in the current manuscript.

Comment 7. P. 22, the authors mention the Runco Ideationnal Behavior Scale as a measure of creative achievement. This scale is not a measure of creative achievement, it is a measure of ideational “habits”, no item of this scale is focused on achievement whatsoever.

Response. We appreciate your comment. We excluded the description of Runco Ideational Behavior Scale as a measure of creative achievement in the current manuscript.

Comment 8. At the very end of the paper, the authors say that future research should clarify the influence of personality on creativity. The literature on this topic is so vast and so much work have been done in this direction that such a comment is simply not acceptable as a way to close the paper!

Response. As you suggested, the end of the last manuscript was not an appropriate discussion about personality and creativity. Thus, the current manuscript excluded the part.

---

## [Editor Report · Decision Letter 3]

8 Aug 2022

Relationships among creativity indices: Creative potential, production, achievement, and beliefs about own creative personality

PONE-D-21-18303R3

Dear Dr. Takagishi,

We’re pleased to inform you that your manuscript has been judged scientifically suitable for publication and will be formally accepted for publication once it meets all outstanding technical requirements.

Kind regards,

Denis Alves Coelho, PhD

Academic Editor

PLOS ONE

---

## [Editor Report · Acceptance letter]

1 Sep 2022

PONE-D-21-18303R3 

Relationships among creativity indices: Creative potential, production, achievement, and beliefs about own creative personality 

Dear Dr. Takagishi:

I'm pleased to inform you that your manuscript has been deemed suitable for publication in PLOS ONE. Congratulations! Your manuscript is now with our production department. 

Kind regards, 

on behalf of

Dr. Denis Alves Coelho 

Academic Editor

PLOS ONE